# On Generalization Across Environments In Multi-Objective Reinforcement Learning

**Jayden Teoh**[1][*]  **Pradeep Varakantham**[1]   **Peter Vamplew**[2]
[1]Singapore Management University    [2]Federation University Australia
[1]{jxteoh.2023,pradeepv}@smu.edu.sg   [2]p.vamplew@federation.edu.au

## Abstract

Real-world sequential decision-making tasks often require balancing trade-offs between multiple conflicting objectives, making Multi-Objective Reinforcement Learning (MORL) an increasingly prominent field of research. Despite recent advances, existing MORL literature has narrowly focused on performance within static environments, neglecting the importance of generalizing across diverse settings. Conversely, existing research on generalization in RL has always assumed scalar rewards, overlooking the inherent multi-objectivity of real-world problems. Generalization in the multi-objective context is fundamentally more challenging, as it requires learning a Pareto set of policies addressing varying preferences across multiple objectives. In this paper, we formalize the concept of generalization in MORL and how it can be evaluated. We then contribute a novel benchmark featuring diverse multi-objective domains with parameterized environment configurations to facilitate future studies in this area. Our baseline evaluations of state-of-the-art MORL algorithms on this benchmark reveals limited generalization capabilities, suggesting significant room for improvement. Our empirical findings also expose limitations in the expressivity of scalar rewards, emphasizing the need for multi-objective specifications to achieve effective generalization. We further analyzed the algorithmic complexities within current MORL approaches that could impede the transfer in performance from the single- to multiple-environment settings. This work fills a critical gap and lays the groundwork for future research that brings together two key areas in reinforcement learning: solving multi-objective decision-making problems and generalizing across diverse environments. We make our code available at https://github.com/JaydenTeoh/MORL-Generalization.

## 1 Introduction

Developing agents capable of generalizing across diverse environments is a central challenge in reinforcement learning (RL) research. While significant progress has been made in studying the generalizability of RL algorithms, these efforts predominantly focus on optimizing a single scalar reward signal (Zhang et al., 2018; Cobbe et al., 2019; Irpan & Song, 2019; Packer et al., 2019; Kirk et al., 2023). Single-objective RL (SORL) overlooks the complexity of real-world problems, which often necessitate trade-offs to be made between multiple conflicting objectives. Reducing these multifaceted considerations to a single scalar reward (objective) obscures critical interactions between the objectives and limits the agent's utility (Vamplew et al., 2022). The field of Multi-Objective Reinforcement Learning (MORL) has sought to address the inherent multi-objective nature of sequential decision-making tasks (Roijers et al., 2013; Hayes et al., 2022). However, the existing body of MORL research has concentrated on optimizing agent performance within *static environments*, neglecting the dimension of generalization across varying situations. Consequently, there exists a significant gap in the RL literature: the intersection of generalization and MORL.

Generalising over multiple scenarios and objectives simultaneously is routinely demanded in many real-world applications, such as healthcare management, autonomous driving, and recommendation

---

[*]Corresponding author.

systems. Consider an autonomous vehicle, which must not only generalize across varied environmental conditions—different weather patterns, lighting, and road surfaces—but also learn optimal trade-offs between competing objectives such as fuel consumption, travel time, passenger's comfort, and safety. Failure to effectively generalize across these environments and objectives would lead to inefficient operation or even catastrophic outcomes. The real world's dynamic nature extends beyond just environmental variability, but also includes evolving goals and utility preferences. An agent optimizing a single scalar reward may exhibit some level of generalization, such as adapting to state variations, but it will struggle to generalize when faced with new goals or reward structures. This is because the agent has only observed its current reward signal, and lacks the basis for adapting its behaviour should the reward signal change. In contrast, a MORL agent learns to consider all dimensions of a vector reward, even those that are not immediately relevant to current goals. This holistic approach to learning allows the agent to adapt swiftly when its utility landscape evolves or when stakeholders' prioritisation over the different objectives shifts. For example, in autonomous driving, a generally capable MORL agent can satisfy unique preferences over objectives for different passengers without the need for retraining. Therefore, developing generally capable multi-objective agents enables not only generalization across *diverse environments*, but also across *dynamic goals and utility functions*—an overlooked aspect in current single-objective RL generalization literature, yet one that is arguably essential for real-world applicability.

The contributions of this paper are as follows: (1) we present formal frameworks for discussing and evaluating generalization in MORL, (2) we introduce the *MORL-Generalization* benchmark, a collection of multi-objective domains with rich environmental variations, (3) we conduct extensive evaluations demonstrating that state-of-the-art (SOTA) algorithms fall short on our benchmark, and (4) we provide post-hoc analyses of possible failure modes in these methods and offer directions for future research. Notably, we open-source our software and release a comprehensive dataset derived from over 1,700 cumulative days of baseline evaluations across multiple SOTA algorithms. These contributions lay the foundation for advancing MORL generalization research, ultimately pushing the boundaries of what RL agents can achieve in complex, real-world environments.

## 2 BACKGROUND

In this section, we introduce MORL and establish the formal notations referenced throughout this paper. A multi-objective sequential decision-making problem can be modeled by a *Multi-Objective Markov Decision Process* (MOMDP; White (1982)) represented by the tuple: $\langle \mathcal{S}, \mathcal{A}, \mathcal{T}, \mathbf{R}, \mu, \gamma \rangle$ with state space $\mathcal{S}$, action space $\mathcal{A}$, transition function $\mathcal{T} : \mathcal{S} \times \mathcal{A} \times \mathcal{S} \to [0, 1]$, initial state distribution $\mu$, and discount factor $\gamma \in [0, 1)$. The key distinction between MOMDPs and standard MDPs lies in the vector-valued reward function $\mathbf{R} : \mathcal{S} \times \mathcal{A} \times \mathcal{S} \to \mathbb{R}^k$, where $k$ is the number of objectives. The goal of a standard RL agent is to maximize its expected long-term discounted sum of rewards, i.e. value function. For a stationary policy $\pi : \mathcal{S} \times \mathcal{A} \to [0, 1]$, the *multi-objective state value function* at state $s \in \mathcal{S}$ is given by

$$\mathbf{V}^\pi(s) := \mathbb{E}_\pi \Big[ \sum_{t=0}^\infty \gamma^t \mathbf{r}_{t+1} | s_0 = s \Big],$$

where $\mathbf{r}_{t+1} = \mathbf{R}(s_t, a_t, s_{t+1})$ is the $k$-dimensional reward vector received at timestep $t + 1$. The expected value vector of $\pi$ under the initial state distribution $\mu$ is defined as $\mathbf{v}^\pi = \mathbb{E}_{s_0 \sim \mu}[\mathbf{V}^\pi(s_0)]$. Since $\mathbf{V}^\pi(s)$ is a vector-valued function, it can only specify a *partial ordering* over the policy space for a given state. Given two policies $\pi$ and $\pi'$. it is possible that $\mathbf{v}_i^\pi > \mathbf{v}_i^{\pi'}$ for objective $i$, but $\mathbf{v}_j^\pi < \mathbf{v}_j^{\pi'}$ for objective $j$. Since there may be no single policy that is optimal across all objectives, MORL requires the agent to learn a *solution set* of policies, each reflecting different trade-offs across objectives. There are two primary approaches to deriving an optimal solution set: the *axiomatic approach* and the *utility-based approach* (Roijers et al., 2013).

The axiomatic approach operates on the axiom that the optimal solution set is the *Pareto set*. This leads us to the concept of *Pareto dominance*. We say a policy $\pi$ *Pareto dominates* (denoted by $\succ_P$) another policy $\pi'$ if its expected value vector is higher or equal across all objectives, that is: $\mathbf{v}^\pi \succ_P \mathbf{v}^{\pi'} \iff (\forall i : v_i^\pi \geq v_i^{\pi'}) \land (\exists i : v_i^\pi > v_i^{\pi'})$. The Pareto Set consists of all nondominated (Pareto optimal) policies:

$$\mathcal{PS}(\Pi) = \{\pi \in \Pi \mid \nexists \pi' \in \Pi, \mathbf{v}^{\pi'} \succ_P \mathbf{v}^\pi\}$$

where $\Pi$ is the set of all possible policies. The image of the Pareto set under the expected value function mapping is known as the *Pareto front*.

On the other hand, the more prevalent utility-based approach assumes each user's preferences over the objectives can be modeled by a utility function that projects the multi-objective value vectors to a scalar utility. In the *multi-policy* setting, utility-based approaches assume user preferences can be represented by a space of weight vectors $\mathbf{w} \in \mathcal{W}$ that parameterize the utility function, i.e. $u : \mathbb{R}^k \times \mathcal{W} \to \mathbb{R}$. These utility functions are usually assumed to be *monotonically increasing* in every objective. This is a natural assumption in accordance with notions of reward—getting more reward for an objective should not decrease a user's utility as long as it does not result in a decrease in reward for another. Linear utility functions, $u(\mathbf{v}, \mathbf{w}) = \mathbf{w}^\mathsf{T}\mathbf{v}$, with weights satisfying $\sum_i \mathbf{w}_i = 1$ and $\mathbf{w}_i \geq 0$, are commonly employed. During training, utility-based agents aim to learn a *coverage set*, $\mathcal{CS} \subset \Pi$, that maximizes the scalar utility for all weights. Formally, this means:

$$\forall \mathbf{w} \in \mathcal{W}, \exists \pi \in \mathcal{CS} \quad \text{s.t.} \quad \mathbf{v}_\mathbf{w}^\pi = \max_{\pi' \in \Pi} \mathbf{v}_\mathbf{w}^{\pi'}.$$

where $\mathbf{v}_\mathbf{w}^\pi$ denotes the *scalar* utility of policy $\pi$ under the utility function $u(\cdot, \mathbf{w})$. Thus, every weight vector $\mathbf{w}$, $\mathcal{CS}$ contains at least one optimal policy that achieves the maximal scalar utility. There are two optimization criteria for which the calculation of $\mathbf{v}_\mathbf{w}^\pi$ are different: *expected scalarized return* (ESR), $\mathbf{v}_\mathbf{w}^\pi = \mathbb{E}_{\pi, s_0 \sim \mu}[u(\sum_{t=0}^\infty \gamma^t \mathbf{r}_t, \mathbf{w})|s_0]$, and *scalarized expected return* (SER), $\mathbf{v}_\mathbf{w}^\pi = u(\mathbb{E}_{\pi, s_0 \sim \mu}[\sum_{t=0}^\infty \gamma^t \mathbf{r}_t|s_0], \mathbf{w})$. When the utility function $u$ is non-linear, these two formulations generally yield different optimal solution sets.

Axiomatic approaches seek to learn the optimal Pareto set and present it to users for selection *a posteriori*. While the Pareto set always contains the policy that maximizes the scalarized value for any monotonically increasing utility function, it can become prohibitively large to learn and to retrieve. In contrast, utility-based approaches narrows the solution set by only focussing on policies that maximize the user's utility. It also simplifies policy selection, as the user's preference, represented by a weight vector $\mathbf{w}$, directly guides the choice of policy during inference. However, this approach requires the users' utility function to be specifiable in closed form and known *a priori*.

## 3 MULTI-OBJECTIVE CONTEXTUAL MARKOV DECISION PROCESS

To formalize the notion of generalization in the context of MORL, we need to start with a way to reason about a collection of multi-objective environments. In single-objective RL (SORL), this is often done using the Contextual MDP (CMDP; Hallak et al. (2015)) framework. As such, we formally define a *Multi-Objective Contextual Markov Decision Process* (MOC-MDP)—an adaptation of the CMDP framework to the multi-objective setting.

**Definition 1** (Multi-Objective Contextual MDP). A MOC-MDP is defined by the tuple

$$\langle \mathcal{C}, \mathcal{S}, \mathcal{A}, \gamma, \boldsymbol{\mathcal{M}} \rangle$$

where $\mathcal{C}$ is the *context space* and $\mathcal{S}$ and $\mathcal{A}$ are the shared state and action spaces across environments respectively. This is similar to a CMDP except $\boldsymbol{\mathcal{M}}$ is a function mapping any $c \in \mathcal{C}$ to a MOMDP, i.e. $\boldsymbol{\mathcal{M}}(c) = \langle \mathcal{S}, \mathcal{A}, \mathcal{T}^c, \mathbf{R}^c, \mu^c, \gamma \rangle$.

The context space, $\mathcal{C}$ defines a set of parameters, each representing a different MOMDP. Intuitively, the context can be viewed as a discrete or continuous parameter specifying the multi-objective environment configuration, such as a seed or a vector controlling the environment dynamics. Each configuration varies in its initial state distribution, transition functions and multi-objective rewards, but share enough common structure across which the MORL agent can generalize. MOC-MDP describes a model where for each context there is a potentially distinct optimal Pareto front. The context *remains constant* within a single episode. Throughout this paper, we will also refer to a particular MOC-MDP as a "domain", and its associated "contexts" as "environments", interchangeably.

We begin by formalizing the generalization objective for axiomatic MORL approaches. The main objective of the axiomatic approach lies in identifying all nondominated vectors across the Pareto front. In the case of a MOC-MDP, since there are different Pareto fronts for each context, to attain optimality in any scalarization function for any context, it would involve a union of policies from Pareto sets across contexts. Collectively, these policies form a *global Pareto set*.

**Definition 2** (Axiomatic Generalization). Given a MOC-MDP with policy space $\Pi$, the generalization problem for axiomatic approaches is to learn a *global Pareto set* $\mathcal{PS}_\mathcal{C}$ given by

$$\mathcal{PS}_\mathcal{C} = \{\pi \in \Pi \mid \exists c \in \mathcal{C}, \nexists \pi' \in \Pi, \ \mathbf{v}_c^{\pi'} \succ_P \mathbf{v}_c^\pi\},$$

where $\mathbf{v}_c^\pi$ is the expected value vector in a context $c$. Thus, the global Pareto set comprises of policies that are nondominated in at least one context, ensuring that all necessary policies for constructing the Pareto fronts in every context are captured.

Recall that in utility-based approaches, each user's preference is modeled by a weight vector, $\mathbf{w} \in \mathcal{W}$, which parameterizes a utility function $u$. Thus, utility-based approaches seek to learn optimal policies for each $\mathbf{w}$. However, in a MOC-MDP, the optimal policies for a given $\mathbf{w}$ may vary across contexts. Therefore, the generalization objective in utility-based approaches is to find policies that maximize the expected utility over the context distribution for each $\mathbf{w}$.

**Definition 3** (Utility-based Generalization). Given a MOC-MDP, for each weight $\mathbf{w} \in \mathcal{W}$, the generalization objective for utility-based approaches is to learn a *generalized coverage set* $\mathcal{CS}_\mathcal{C}$ given by

$$\forall \mathbf{w} \in \mathcal{W}, \exists \pi \in \mathcal{CS}_\mathcal{C} \quad \text{s.t.} \quad \mathbb{E}_{c \sim p(c)}[\mathbf{v}_{c,\mathbf{w}}^\pi] = \max_{\pi' \in \Pi} \mathbb{E}_{c \sim p(c)}[\mathbf{v}_{c,\mathbf{w}}^{\pi'}],$$

where $p(c)$ is the context distribution and $\mathbf{v}_{c,\mathbf{w}}^\pi$ denotes the scalar utility of $\pi$ for context $c$ under the utility function $u(\cdot, \mathbf{w})$, computed using either the ESR or SER criterion. In this framework, each policy must generalize across contexts for a specific $\mathbf{w}$. This structure retains the key advantage of utility-based methods—allowing users to easily select their preferred policy by indicating $\mathbf{w}$ at inference time, independent of the context.

When the context is fully observable, one can simply augment the state-space with the sampled context $c$ itself, e.g. $s' = concat(s, c)$. This effectively reduces the MOC-MDP to a "universal" MOMDP that unifies the context-dependent components, i.e. $\mathcal{T}^c, \mathbf{R}^c, \mu^c$, into a single model. In this setting, the global Pareto set $\mathcal{PS}_\mathcal{C}$ and the generalized coverage set $\mathcal{CS}_\mathcal{C}$ correspond directly to the Pareto set $\mathcal{PS}$ and coverage set $\mathcal{CS}$ of the "universal" MOMDP, respectively.

In settings where the context is hidden or only partially observable at test time, both axiomatic and utility-based generalization methods must infer the underlying context to achieve optimality. That is, the agent must learn a posterior over the context space, i.e. $p(c|\mathcal{H})$, where $\mathcal{H}$ represents observable information in the test environment such as state-action history and rewards. This process is analogous to solving a *partially-observable* MOMDP, in which the agent forms and updates a belief over the current context to determine the optimal policy. Note here that we do not impose restrictions on the nature of the policies learned in $\mathcal{PS}_\mathcal{C}$ and $\mathcal{CS}_\mathcal{C}$. The policies may be either Markovian or non-Markovian, depending on whether context inference is decoupled from the policy.

## 4 EMPIRICAL EVALUATION OF MORL GENERALIZATION PERFORMANCE

Let $\mathcal{C}_{\text{eval}} = \{c_1, c_2, \ldots, c_n\}$ represent a set of independent evaluation contexts. Measuring an agent's generalization performance in SORL is straightforward: the larger the reward value across $\mathcal{C}_{\text{eval}}$, the better. In MORL, however, agents produce an *approximate Pareto front* comprising multiple value vectors for each $c \in \mathcal{C}_{\text{eval}}$, and translating the quality of this Pareto front into a scalar metric that captures generalization performance is non-trivial.

In what follows, we propose an axiomatic-based evaluation metric called the *Normalized Hypervolume Generalization Ratio* (NHGR) to enable fairer assessments of generalization performance in MORL. Our discussions and evaluations focus primarily on this axiomatic-based metric because specifying a meaningful user utility function *a priori* can often be infeasible in reinforcement learning benchmark tasks, including those introduced in Section 5. For completeness, we also introduce a utility-based evaluation metric, the *Expected Utility Generalization Ratio* (EUGR) in the appendix. The EUGR is useful in scenarios where the MORL agent must generalize across multiple environments with well-defined user utility functions. Both metrics assume that an optimal Pareto front is available for each evaluation context. In practice, when the true front is unavailable, it can be approximated by aggregating nondominated value vectors from specialist agents trained independently

in each context. For extended discussions on the benefits of NHGR, evaluations using EUGR and other metrics, we refer motivated readers to Appendix D.

The Normalized Hypervolume Generalization Ratio (NHGR) is an extension of the widely used *Hypervolume* (HV) indicator (Zitzler & Thiele, 1998). The HV is a standard metric for evaluating the quality of approximate Pareto fronts in single-context MORL, measuring the volume in objective space covered by a Pareto front relative to a reference point. However, the HV metric is not scale-invariant and tends to bias evaluations toward objectives with larger magnitudes. While normalization has been explored in multi-objective optimization (MOO) (Deb & Kalyanmoy, 2001), it has been largely overlooked in MORL literature. Therefore, before we introduce the NHGR, we briefly introduce the *Normalized Hypervolume* ($\text{HV}_{\text{norm}}$).

Consider a $k$-objective domain and let $\mathcal{F}_c^* \subset \mathbb{R}^k$ denote the optimal Pareto front for an evaluation context $c \in \mathcal{C}_{\text{eval}}$. We define the *element-wise* minimum and maximum boundary value vectors of $\mathcal{F}_c^*$ as $\mathbf{v}_c^{\min}$ and $\mathbf{v}_c^{\max}$ respectively. We refer to a MORL agent trained to generalize across multiple multi-objective contexts as the *generalist*. Our goal is to evaluate the performance of this generalist. Let $\tilde{\mathcal{F}}_c \subset \mathbb{R}^k$ denote the approximate Pareto front produced by the generalist for context $c$. The Normalized Hypervolume is defined as

$$\text{HV}_{\text{norm}}(\tilde{\mathcal{F}}_c) = \lambda_k \left( \bigcup_{\mathbf{v}^\pi \in \tilde{\mathcal{F}}_c} \left[ \frac{\mathbf{v}^\pi - \mathbf{v}_c^{\min}}{\mathbf{v}_c^{\max} - \mathbf{v}_c^{\min}}, \mathbf{0} \right] \right),$$

where $\lambda_k$ is the $k$-dimensional Lebesgue measure (Lebesgue, 1902). Since the objectives are normalized, we can use the origin $\mathbf{0}$ as the reference point. The use of $\text{HV}_{\text{norm}}$ also enhances interpretability as it is bounded within 0 and the hypervolume of the unit hypercube (which is 1). Using $\text{HV}_{\text{norm}}$, we can then introduce the NHGR metric:

**Definition 4** (Normalized Hypervolume Generalization Ratio). The NHGR for the generalist in context $c$ is defined as:

$$\text{NHGR}(\tilde{\mathcal{F}}_c, \mathcal{F}_c^*) = \frac{\text{HV}_{\text{norm}}(\tilde{\mathcal{F}}_c)}{\text{HV}_{\text{norm}}(\mathcal{F}_c^*)}.$$

NHGR measures the ratio of normalized hypervolume between the generalist's approximate Pareto front and the optimal Pareto front. NHGR draws similarities to the *Hyperarea Ratio* (Veldhuizen & Allen, 1999) metric in MOO literature but additionally employs $\text{HV}_{\text{norm}}$ to ensure scale-invariance. Fig. 1 illustrates the NHGR metric for a biobjective domain. Intuitively, a truly generalizable MORL agent should achieve optimal performance in every context, which corresponds to attaining an NHGR of 1 across all contexts.

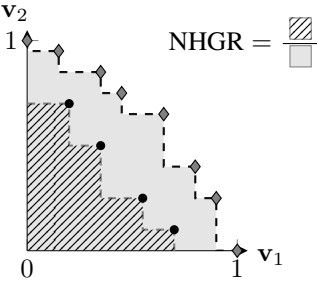

Figure 1: NHGR visualized as the ratio between the hypervolume of the normalized generalist's and the optimal Pareto fronts (dashed and shaded areas).

## 5 MORL GENERALIZATION BENCHMARK

We now introduce the *MORL-Generalization* benchmark, a diverse collection of multi-objective domains with rich environmental variations to facilitate future research on generalization in MORL. We adapted existing domains from MO-Gymnasium (Felten et al., 2023), a multi-objective extension of the Gymnasium library (Towers et al., 2024; Brockman et al., 2016), and introduced new ones, each with expressive parameters controlling environmental variations. Kirk et al. (2023) identified four key types of domain variations for studying generalization: 1) state-space variation (S), which alters the initial state distribution, 2) dynamics variation (D), which alters the transition function, 3) visual variation (O), which impacts the observation function, and 4) reward function variation (R). This benchmark primarily focuses on state-space and dynamics variations. Observation variations do not alter the underlying MOMDP structure (Du et al., 2019). Hence, they provide limited insights into the multi-objective decision-making capabilities of the agent since the optimal Pareto front across variations remain isomorphic. Reward variations are often introduced through multiple goals. Multiple goals can naturally be modeled as multiple objectives by treating each goal as a

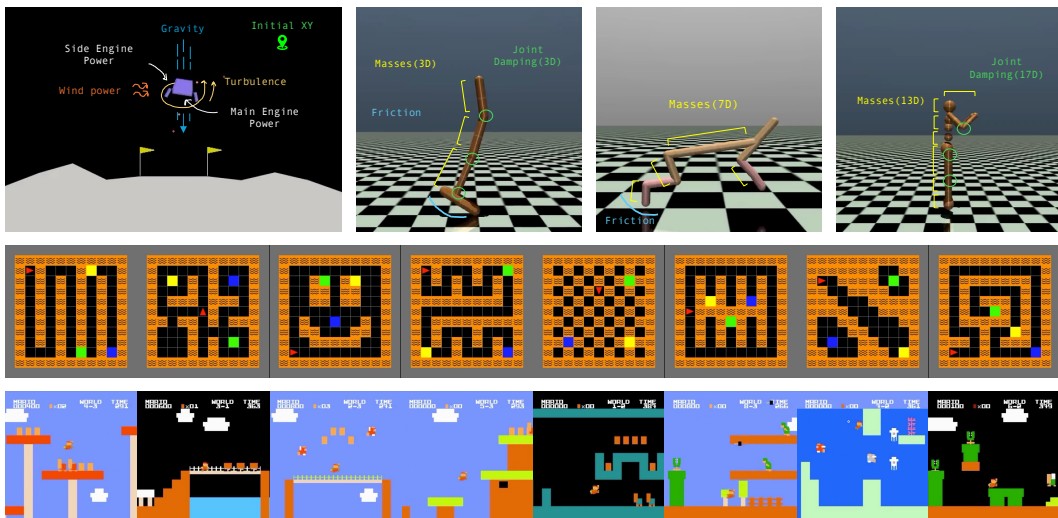

Figure 2: Domains in the MORL Generalization benchmark. Top row from left to right: 1) MO-LunarLander, 2) MO-Hopper, 3) MO-Cheetah, 4) MO-Humanoid. Middle row: MO-LavaGrid (8 handcrafted evaluation environments). Bottom row: MO-SuperMarioBros (8 out of 32 stages).

conflicting objective (Sener & Koltun, 2018), which means MORL inherently involves learning to adapt to reward function variations. Nevertheless, we provided a novel maze domain that explicitly segregates the goals and multiple objectives, for completeness. Fig. 2 visualizes the domains provided in the MORL-Generalization benchmark with annotations for their environment parameters, where applicable. We provide detailed descriptions of each benchmark domain in Appendix F.1.

When designing a generalization benchmark for MORL, it is important to keep in mind of *The Principle of Unchanged Optimality* (Irpan & Song, 2019). This principle asserts that, for a domain to support generalization, it should provide all necessary information such that a policy optimal in every context can exist. We discuss how we uphold this important principle within Appendix B.

## 6 EXPERIMENTS

In this section, we evaluate state-of-the-art MORL algorithms on the newly-developed benchmark to establish baseline expectations for their generalization capabilities. The implementations of these algorithms are adapted from Felten et al. (2023). Specifically, the algorithms evaluated are CAPQL (Lu et al., 2023), Envelope (Yang et al., 2019), GPI-LS (Alegre et al., 2023), PCN (Reymond et al., 2022), PGMORL (Xu et al., 2020), and MORL/D SB (Felten et al., 2024). We also include the model-based extension of GPI-LS, i.e. GPI-PD, and the weight adaptation variant of MORL/D SB, i.e. MORL/D SB+PSA. Additionally, we include the SAC (Haarnoja et al., 2018) algorithm trained with a single objective/utility function in our evaluations to verify that the objectives are not so highly correlated that a single-objective agent could also achieve high performance across multiple objectives. In total, we evaluate 8 MORL algorithms across 6 domains using 5 seeds each. These established baseline performances are open-sourced via *Weights and Biases* (Biewald, 2020), facilitating future research and saving computational resources.

Note that Envelope is restricted to discrete-action domains, while CAPQL and PGMORL apply only to continuous-action domains. MORL/D SB and MORL/D SB+PSA were excluded from MO-SuperMarioBros evaluations due to the high memory demands of convolutional neural networks, which are incompatible with their evolutionary approach. GPI-PD was omitted from MO-HalfCheetah, MO-Humanoid, MO-SuperMarioBros, and MO-LavaGrid, as the large state spaces in these domains cause its dynamics model to degrade rapidly, resulting in zero simulated transitions and effectively reducing it to its model-free counterpart, GPI-LS.

*Domain Randomization* (DR) is an efficient method to expose the agent to a wide range of environments during training by uniformly sampling from the environment parameter space. It has found

success in deep RL even for complex visual domains and real-world robotic control (Tobin et al., 2017; Peng et al., 2018). We utilise DR for all our experiments by randomizing the environment parameters after every training episode. This also enables us to standardise the presentation of environments across algorithms via the RNG seed, and evaluate the algorithms solely for their generalization capabilities. At each evaluation time step, each algorithm is assessed over 100 episodes [1] across a set of environment configurations. Whenever possible, these configurations are chosen using the boundary values of environment parameter ranges to ensure diverse evaluation environments and behaviors. For MORL algorithms using linear scalarization, a weight vector is uniformly sampled from the unit simplex for each evaluation episode. We aggregate the NHGR performance across all evaluation environments for each domain and report results in terms of inter-quartile mean (IQM) and optimality gap using the `rliable` library (Agarwal et al., 2021), which helps account for statistical uncertainty prevalent in deep RL. IQM focuses on the middle 50% of combined runs, discarding the bottom and top 25%. Optimality gap captures the amount by which the algorithm fails to meet a desirable target, i.e. when NHGR=1. The evaluation environment configurations and other experiment setups are detailed in Section F of the appendix for reproducibility.

## 6.1 MORL GENERALIZATION RESULTS

The baseline results using NHGR reveal significant performance gaps in the generalist agents across various domains, highlighting the benchmark's potential to serve as a foundational benchmark for future research in MORL generalization.

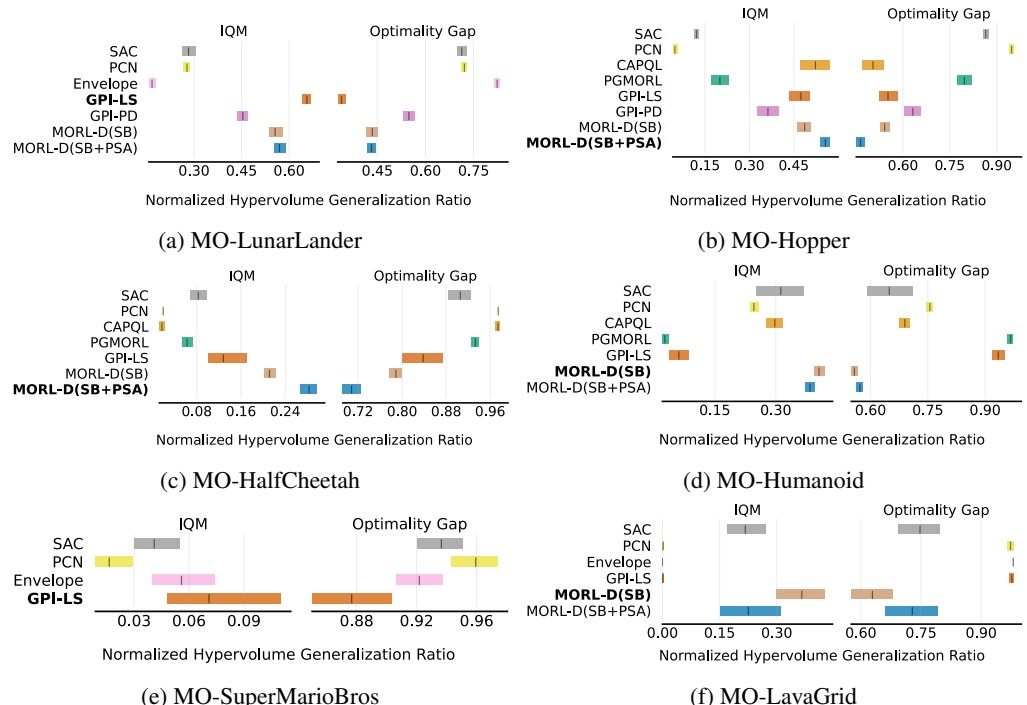

Figure 3: Aggregate NHGR performance in all domains of the benchmark. Each algorithm is evaluated across 5 independent seeds and several evaluation environment configurations. Higher IQM and lower optimality gap scores are better. The best algorithm for each domain is bolded.

**MO-LunarLander** MO-LunarLander is one of the simplest domains in the benchmark, featuring a low-dimensional observation space and discrete action space. The best-performing algorithm, GPI-LS, achieved an IQM NHGR score of 0.66 with an optimality gap of 33.9% (see Fig. 3a). Given its simplicity, we recommend using MO-LunarLander as a starting point for evaluating new algorithms. Additionally, our software includes a continuous-action variant of this domain, which may reveal larger NHGR optimality gaps and provide greater opportunities for improvement.

---

[1]In MO-SuperMarioBros, 32 evaluation episodes are used to keep runtime under 72 hours.

**MuJoCo-based Domains** The challenge of generalization in MORL becomes more pronounced in the MuJoCo-based (Todorov et al., 2012) domains. Across the 3 domains, a wider spread in performances and noticeably lower performance ceilings are observed. In the MO-Hopper domain, the leading algorithm, MORL/D SB+PSA, managed to reach an IQM NHGR score of only 0.56 and optimality gap of 46.4% (see Fig. 3b). This gap widens in higher-dimensional domains like MO-HalfCheetah (Fig. 3c) and MO-Humanoid (Fig. 3d), where the top algorithms, MORL/D SB+PSA and MORL/D SB, achieve IQM NHGR scores of just 0.28 and 0.41, respectively. These wide optimality gaps, combined with the strong relevance to real-world robotic control tasks, suggest that these domains may serve as enduring benchmarks for studying generalization in MORL.

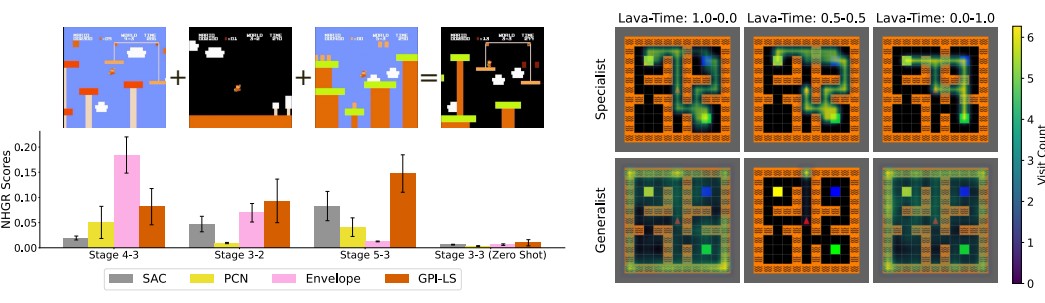

(a) MO-SuperMarioBros Performances      (b) Heatmap of visited tiles

Figure 4: (a) MO-SuperMarioBros performances on 4 stages. Stage 3-3 in the rightmost column shares visual similarities with the other stages so it is excluded from training to evaluate for zero-shot generalization. (b) Heatmap of visited tiles for a specialist and generalist in the MO-LavaGrid "Room" environment. Each column's title shows the conditioned linear weights for the lava and time penalty objectives.

**MO-SuperMarioBros** Fig. 3e presents the NHGR performance of 3 discrete MORL algorithms and SAC on MO-SuperMarioBros. The leading algorithm, GPI-LS, achieved an IQM NHGR score of only 0.07 and optimality gap of 87.7%. We also conducted a *zero-shot generalization* experiment by excluding stage 3-3 from the training distribution. This stage shares a combination of visual features with stages 3-2, 4-3, and 5-3, allowing us to test if an agent has learned generalizable behaviors over the pixel space or merely memorized stage-specific sequences. The results in Fig. 4a show negligible NHGR performances in the zero-shot environment (stage 3-3), suggesting the latter.

**MO-LavaGrid** Fig. 3f shows the performance of five discrete MORL algorithms on MO-LavaGrid, with MORL/D SB achieving the highest IQM NHGR score of 0.37, though still far from optimal. To analyze this, we recorded trajectories for a generalist (MORL/D SB) and a specialist (GPI-LS) in the "Room" environment, both using linear scalarization. Fig. 4b presents heatmaps of tile visit counts when conditioned on three different weightings of lava and time penalties. While the specialist consistently follows optimal routes, the generalist exhibits random walks overlapping with the three goals. This likely explains MORL/D SB's nonzero NHGR performance across environments but significant optimality gap, as it incurs high penalties from inefficient navigation.

In summary, the generalization performance of the current MORL algorithms leaves much to be desired. This outcome is not surprising, as these experiments were aimed to provide a baseline understanding of existing methods without any tailored interventions to enhance generalization yet. Despite not attaining the top performance in every domain, MORL/D SB and MORL/D SB+PSA, demonstrated the most consistent results overall. Future research aiming to improve MORL generalizability can consider building upon these algorithms for more reliable testing.

### 6.2 SCALAR REWARD IS NOT ENOUGH FOR RL GENERALIZATION

Real-world problems are often multi-objective. In fact, many popular SORL benchmarks are naturally multi-objective but are simplified using hidden scalarization functions. For instance, the original Hopper domain combines forward velocity ($v_x$), control cost ($c$), and a bonus for not falling ($h$) into a scalar reward: $1.5v_x + 0.001c + h$. In contrast, MORL approaches treat these as independent objectives and may even introduce additional ones, such as torso height. One might argue that if a

stakeholder's sole goal is to maximize the agent's forward movement, MORL approaches that seek to learn tradeoffs across multiple objectives would be redundant in RL generalization. However, our empirical results challenge this assumption.

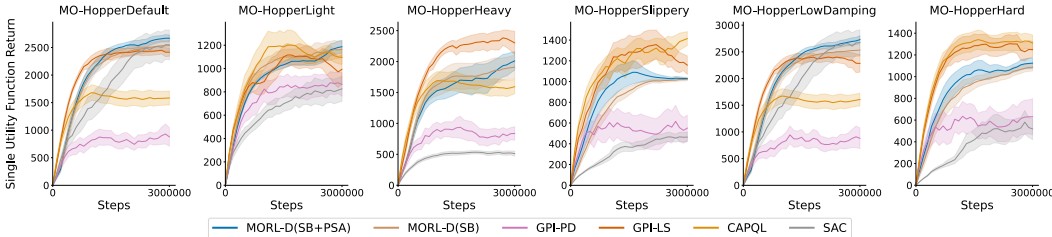

Figure 5: Single-objective return on 6 MO-Hopper testing environments during training. Each curve is measured across 5 seeds (mean and standard error).

Let $f_{\text{SORL}}$ denote the fixed scalar reward function that SORL agents are trained to optimise during generalization. Throughout the generalization training horizon of the MORL algorithms in Section 6.1, we sampled solution vectors across their Pareto fronts, scalarized them using $f_{\text{SORL}}$, and recorded the highest scalar reward in each evaluation environment. For the SORL agent, which already specializes on $f_{\text{SORL}}$, we allow it as many runs as solution vectors sampled from the MORL agents, and took the best result. Our results reveal that when trained on the same generalization procedure, MORL algorithms can outperform the SORL agent on its own specialized reward function $f_{\text{SORL}}$. Fig. 5 shows several MORL algorithms surpassing the SAC agent by large margins in $f_{\text{SORL}}$ return across six distinct test environments during generalization training in the MO-Hopper domain. Note that CAPQL is a multi-objective variant of SAC, while MORL/D SB and MORL/D SB+PSA is population-based approach of SAC. All SAC-based implementations are from the same library, CleanRL (Huang et al., 2022), making these results fair. Similar findings are observed in other domains (see Section D.3 in appendix), where leading MORL algorithms could outperform or achieved parity with SAC on $f_{\text{SORL}}$ performance.

Fig. 6 presents snapshots from the highest $f_{\text{SORL}}$ episodes of the MORL/D SB+PSA agent on three MO-Hopper environments. Since MORL/D SB+PSA is a linear utility-based approach, the table in Fig. 6 provides the linear weight vectors which the agent conditioned on. In the *Default* environment, the agent placed a higher weight on forward velocity, causing it to lean forward and cover more distance. In the *Slippery* environment, where low friction makes leaning forward dangerous, the agent balances forward velocity with torso height, maintaining an upright posture to prevent slipping. In the *Hard* environment, which features a slippery floor, unbalanced body masses, and low joint damping, the agent maximized $f_{\text{SORL}}$ by emphasizing torso height and minimizing control cost, which help maintain stability and avoid abrupt movements. In contrast, the single-objective SAC agent learned only a single behavior to generalise across all environments, causing it to fail in dire conditions.

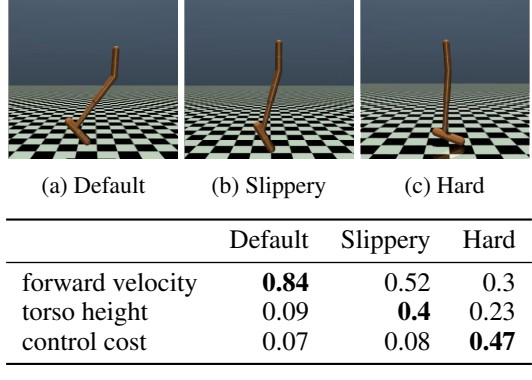

| | Default | Slippery | Hard |
|---|---|---|---|
| forward velocity | **0.84** | 0.52 | 0.3 |
| torso height | 0.09 | **0.4** | 0.23 |
| control cost | 0.07 | 0.08 | **0.47** |

Figure 6: Screenshots of MORL/D SB+PSA agent's behavior in different MO-Hopper environments and the corresponding linear weights.

The single-objective approach to RL generalization is heavily reliant on *reward engineering*, i.e. finding an optimal scalar reward signal through trial-and-error search of scalarization functions (Sutton & Barto (2018), Chapter 17.4). However, our observations suggest that there may be no universal scalarization function capable of optimizing performance across all environments. Each environment demands distinct behaviors from the agent, even for a fixed goal like moving forward. Consequently, *a priori* scalarization in SORL limits the agent's adaptability to environmental changes. In contrast, generalization with MORL approaches circumvents the reward engineering problem by considering all dimensions of a vector reward independently, even those not immedi-

ately relevant to current goals. This flexibility allows agents to learn diverse behaviors that address different trade-offs among objectives. Stakeholders can then select policies from the agent's solution set that best match their utility preferences for any given environment, enhancing the adaptability of MORL agents in generalization tasks. These observations align with recent studies that challenge the expressivity of scalar rewards and advocate for the adoption of multi-objective reward formulations (Vamplew et al., 2022; Skalse & Abate, 2024; Subramani et al., 2024).

# 7 RELATED WORK

**Multi-Objective Contextual Multi-Armed Bandits:** *Multi-Objective Contextual Multi-Armed Bandits* (MOC-MAB; Tekin & Turgay (2017); Turgay et al. (2018)) are a context-dependent, multi-objective extension of the classic *Multi-Arm Bandit* (MAB) problem. In MOC-MAB, at each decision point, the agent observes a context and selects an action (arm) to maximize a vector of immediate rewards corresponding to different objectives. While MOC-MAB provides valuable insights into handling contexts and balancing multiple objectives simultaneously, it fundamentally differs from the MOC-MDP framework. Specifically, MOC-MAB does not address the state-transitions and sequential decision-making inherent in MORL. Our work extends beyond the MOC-MAB setting by focusing on the generalization of RL agents in a context-dependent, multi-objective environment—a problem that, to our knowledge, has not been previously explored in the literature. However, bandit analysis often forms the foundations of progress in RL, so we implore future work to look into the MOC-MAB framework for inspiration on improving generalization in MORL.

**Multi-Task Learning:** *Multi-Task Learning* (MTL; Caruana (1998)) and *Multi-Task Reinforcement Learning* (MTRL; Tanaka & Yamamura (2003)) aim to improve learning efficiency and performance by leveraging shared representations across multiple tasks. Reinforcement learning based on Contextual MDP (CMDP) is closely related to MTRL but involves a parameterized variable, termed the context, which allows for a more unified modeling of tasks within a single framework. However, both MTRL and CMDP have predominantly been studied in the single-objective setting, focusing on maximizing a scalar reward function. Sener & Koltun (2018) framed MTL as a MOO problem by treating different tasks as conflicting objectives. While this perspective introduces multi-objectivity into MTL, it primarily addresses trade-offs between tasks rather than scenarios where each task involves multiple objectives. In the optimization domain, the *Multi-Objective Multifactorial Optimization* paradigm (Gupta et al., 2017) considers multitasking across multiple multi-objective problems by leveraging shared evolutionary operators to solve them simultaneously.

Despite these advancements, there is a notable gap in the literature regarding the simultaneous consideration of multi-objectivity and generalization across contexts (i.e. environments or tasks) in reinforcement learning. To the best of our knowledge, this is the first study to systematically explore generalization in MORL and highlight unique difficulties within this combined setting.

# 8 DISCUSSION AND CONCLUSION

Developing reinforcement learning agents for real-world tasks necessitates not only generalization across diverse environments, but also across multiple objectives. By formally introducing a framework for discussing and evaluating generalization in MORL, we bridge a crucial gap between RL generalization and multi-objective decision-making. To measure progress in this area, we contributed a novel benchmark to facilitate rigorous investigations into MORL generalization. The extensive baseline evaluations of state-of-the-art MORL algorithms on the benchmark also highlight significant room for future research to improve upon. We encourage readers to look at Section A of the appendix, where we analyzed algorithmic failure modes in current MORL approaches, offering insights into their poor generalization performance. Extended discussions of MORL generalization and future research directions are also provided in the appendix. Moreover, we have open-sourced our software, alongside the raw dataset derived from our baseline evaluations. These contributions would streamline future investigations into MORL generalization. We hope this paper spurs greater recognition of the importance of multi-objective reward structures for RL generalization. Ultimately, by unifying these two fields, this paper lays the groundwork for advancing RL agents capable of tackling the complexities of real-world, multi-objective scenarios.

## ACKNOWLEDGMENTS

This research/project is supported by the National Research Foundation Singapore and DSO National Laboratories under the AI Singapore Programme (AISG Award No: AISG2-RP-2020-017).

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

## A    ANALYSIS OF FAILURE MODES IN MORL APPROACHES

In this section, we seek a deeper understanding on failure modes within the current MORL algorithms that can hinder generalization. We caution readers looking to further MORL generalization to be wary of them and encourage exploration to solve these failure modes. Note that we will only discuss challenges that are unique to generalization within MORL, and problems pertaining to the broader RL generalization literature are excluded.

**Pareto Archival Methods**    MORL methods often maintain a *Pareto archive*—a set of nondominated policies discovered during training. This archive is constantly updated by comparing the value vector of new policies with old ones, and discarding the dominated ones. This archive can then aid the agent's search process within the objective space, or be used as solutions during test time. This technique is commonly used in multi-objective evolutionary algorithms like PGMORL and MORL/D. Similarly, GPI-LS and GPI-PD track a finite convex subset of the Pareto front where dominance is defined only for linear utility functions. However, when extending these methods to a MOC-MDP—where each context has its own optimal Pareto front—current archiving mechanisms can lead to suboptimal outcomes. Most MORL literature assumes a static environment, so existing Pareto archival mechanisms are not designed to handle context variability in MOC-MDPs. As a result, the archive overrepresents policies that perform well in a narrow set of contexts with higher reward scales or lower difficulty, while discarding those optimal for more challenging or less rewarding contexts. This has severe implications as it will cause the agent to converge to a *maximax strategy*, adopting policies that are only optimized to yield the best of the best possible outcomes during test time, and results in poor generalization across the entire range of contexts in the MOC-MDP.

**Reliance on Linear Scalarization**    The convexity of the induced value functions' range determines if MORL algorithms relying on linear scalarization (LS), are capable of finding all policies corresponding to the optimal Pareto front (Vamplew et al., 2008; Roijers et al., 2013). Lu et al. (2023) showed that in the static-environment setting, the induced value functions' range of stochastic stationary policies in a MOMDP is convex, which means LS is not a bottleneck for approximating the Pareto front. If the learning objective is to maximize the expected (average) multi-objective value function across contexts in a context-agnostic manner, the results from Lu et al. (2023) can be directly applied to show that the range of expected value vectors in a MOC-MDP remains convex. This follows trivially from the linearity of the expectation operator, which preserves convexity. However, if our maximization objective is the recovery of the optimal Pareto Front or coverage set across all contexts, the policies that the agent learn may need to be non-stationary and/or non-Markovian (further discussed in Section B). Prima facie, in cases where the policies exhibits nonlinear dependence on state-action history, methods relying solely on LS would be insufficient to identify all globally Pareto-optimal policies in a MOC-MDP. We encourage future research to further investigate the viability of LS in MORL generalization and to explore alternative approaches capable of approximating the Pareto front without convexity assumptions, such as nonlinear scalarization methods or evolutionary algorithms.

**Value Function Interference**    Within state-of-the-art MORL, many approaches extend value-based scalar RL algorithms such as Q-learning or Deep Q-Networks to handle vector rewards. If the utility function allows actions with widely differing vector outcomes to map to the same utility value, then the vector value function learned for earlier states may be inconsistent with the actual optimal policy (Vamplew et al., 2024). This problem is particularly likely to arise in environments which are stochastic or partially-observable. We note that for MOC-MDPs, the dynamics and rewards observed by the agent may appear to be stochastic even if the underlying MDPs are deterministic, due to the influence of the hidden context variables. Therefore, value function interference may pose a particular problem when naively applying value-based MORL algorithms to MOC-MDPs. We note that if the utility function is linear then value interference does not impact on selecting the optimal action, hence there is an implicit tension between this failure mode, and the issues of reliance on linear scalarisation raised in the previous paragraph.

## B    Principle of Unchanged Pareto Optimality

When constructing a benchmark in reinforcement learning, it is essential that the domain adheres to *The Principle of Unchanged Optimality* (Irpan & Song, 2019)—a fundamental yet underappreciated principle. This principle states that, for a domain to support true generalization, it must provide all necessary information such that an optimal policy exists in every context. In the MOC-MDP framework, *The Principle of Unchanged (Pareto) Optimality* implies the existence of globally Pareto optimal policies, $\pi^*$, such that:

$$\forall c \in \mathcal{C} : \quad \pi^* \in \mathcal{PS}_c(\Pi),$$

where $\Pi$ is the set of feasible policies and $\mathcal{PS}_c(\Pi)$ denotes the Pareto set containing nondominated policies for a given $c \in \mathcal{C}$. This principle has significant theoretical implications. When the unchanged optimality principle is disregarded, the benchmark can become a proxy measure of the memorization capability (Zhang et al., 2018) of the MORL agents, instead of generalization. Moreover, if the principle is violated, generalist agents would never achieve an aggregated NHGR score of 1, since they would fail to recover Pareto optimality across all contexts.

This section examines how our *MORL-Generalization* benchmark upholds The Principle of Unchanged Pareto Optimality, thereby validating our baseline evaluations. Note that each context in our benchmark varies in initial state distribution, transition dynamics, and multi-objective reward function. When the context is fully observable, the agent can simply include the context in its state representation, enabling it to learn "universal" policies that adapt across contexts. In scenarios where the context is hidden, the agent must infer it from its observations. Therefore, to respect The Principle of Unchanged Pareto Optimality, our benchmark is designed to ensure that *necessary information about the context* can be recovered from the agent's observations in the proposed domains.

In MO-SuperMarioBros, despite visual similarities across levels, each observation provides sufficient information to determine the optimal action at every time step. For example, the locations of coins, enemies, and bricks are clearly visible. Moreover, since there are only a finite number of stages (32), the agent can deduce its current stage directly from its observations with enough training. Similarly, in MO-LavaGrid, the complete layout of lava and goals, along with the agent's position and orientation, is fully observable at each time step. Furthermore, as described later in Section F, we concatenate the reward weights for each goal with the agent's observation, ensuring that the current reward function is explicitly provided.

For the continuous control domains like MO-Hopper, MO-HalfCheetah, and MO-Humanoid, context variations arise from changes in dynamics (e.g., gravity, friction), yet the agent's observations typically include only joint positions and velocities. Consequently, optimal actions cannot be inferred from a single time step. A similar limitation exists in the discrete domain MO-LunarLander, where the observations are typically restricted to orientation and velocity. The environment dynamics is, however, inferable when the agent considers its state-action history. Prior work has shown that history-based policies are effective in domains with changing dynamics (Yu et al., 2017; Peng et al., 2018; Tiboni et al., 2024). Therefore, we adopt the standard approach of augmenting the state with a fixed-length history of past state-action pairs. In our main experiments for MO-Hopper, MO-HalfCheetah, MO-Humanoid, and MO-LunarLander, we use a history length of 2 so that the observed state at time $t$ is a vector of the form: $(s_{t-2}, a_{t-2}, s_{t-1}, a_{t-1}, s_t)$. For time steps before 2, we repeat the initial state and pad missing actions with zeros.

We recommend that future researchers verify that The Principle of Unchanged Pareto Optimality is upheld when using our software to study MORL generalization. This is crucial for establishing benchmarks that accurately assess an algorithm's capacity to generalize across diverse multi-objective domains.

## C    Future Work and Limitations

Our extensive evaluations of current MORL algorithms on the introduced benchmark, have highlighted a significant gap in generalization capabilities. The suboptimal performance observed underscores the necessity for innovative approaches to enhance MORL generalization.

A promising direction for future research involves adapting established methods from single-objective RL generalization to the multi-objective context. Techniques such as regularization techniques (Cobbe et al., 2019; Ahmed et al., 2019; Li et al., 2019; Igl et al., 2019; Eysenbach et al., 2021), incorporating inductive biases (Tang et al., 2020; Raileanu & Fergus, 2021; Higgins et al., 2018), and curriculum learning methods (Narvekar et al., 2020; Jiang et al., 2021; Teoh et al., 2024) have demonstrated efficacy in single-objective settings and could be tailored to address the complexities inherent in MORL.

Beyond adapting existing methods, there is a need to develop specialized techniques targetted towards MORL. As highlighted in Section A, many current methods rely heavily on linear scalarization, potentially limiting the generalization potential of MORL agents. Exploring alternative approaches that move beyond this constraint can be a promising direction. Recently, evolutionary methods such as those proposed by Xu et al. (2020) and Felten et al. (2024) have been introduced, but they remain underexplored in the MORL literature and warrant further investigation to enhance generalization. Effective exploration is critical for generalization in RL (Jiang et al., 2023). Thus, approaches like Vamplew et al. (2017), which incorporate exploration techniques from single-objective RL into the MORL framework could be of interest to future research. In many real-world scenarios, agents operate under partially observable contexts, such as the dynamics of the environment, as just described in Section B. Developing MORL algorithms capable of adapting to partially-observable contexts is crucial for generalization. Therefore, future work can take inspiration from the POMDP or model-based RL literature to develop methods capable of inferring the hidden context and recovering Pareto optimality.

**Limitations** Our *MORL-Generalization* benchmark is built using Gymnasium's API, which has become the standard interface for reinforcement learning due to its accessibility. However, Gymnasium relies on CPU-based policy rollouts, leading to significant computational bottlenecks caused by frequent GPU-CPU data transfers. This limitation is particularly problematic for MORL algorithms, which generally require longer training times. Most of our experiments complete within 1–2 days, with all runs kept under five days on a single NVIDIA RTX A5000 GPU and a 48-core AMD EPYC 7643 CPU. Given these computational constraints, exhaustive hyperparameter tuning is impractical. Instead, we tune hyperparameters within the neighborhood of those already validated by the *MORL-Baselines* (Felten et al., 2023) library in single-environment settings, from which we also derive the implementations of the baseline algorithms. Future work on extending the *MORL-Generalization* domains to run directly on GPU, enabling fully end-to-end MORL training, would significantly benefit the community.

## D    EXTENDED METRIC DISCUSSIONS AND RESULTS

In this section, we extend our discussions on the benefits of using the NHGR metric. We also provided a utility-based evaluation metric for measuring generalization performance. However, as a pioneering effort, our aim is not to prescribe a single definitive metric for evaluating generalization in future research, but rather to establish well-justified and flexible options. The choice of performance metric has long been debated in multi-objective optimization, as assessing the quality of Pareto fronts is inherently more complex than single-objective evaluations—no single metric can fully capture all aspects of performance. To provide a comprehensive assessment, we report results using multiple metrics, including hypervolume, expected utility metric (EUM), NHGR, and EUGR, as shown in Table 2. Additionally, while not explicitly included in our results, our software supports other evaluation metrics such as cardinality and sparsity for further analysis.

### D.1    EXTENDED DISCUSSIONS ON BENEFITS OF NHGR

In Section 4, we introduced the *Normalized Hypervolume* ($HV_{norm}$) to ensure equal weighting across objectives in hypervolume calculations. This is achieved by normalizing reward scales using the minimum and maximum values derived from the optimal Pareto front (or an approximation thereof) before computing hypervolume. Ensuring equal weightage across objectives when measuring generalization performance is important. This is because, unlike in single-objective RL, MORL requires agents to generalize across both environments and objectives, making equal weighting crucial to ensure that they learn all trade-offs across objectives effectively. Additionally, $HV_{norm}$ removes the

dependence on an arbitrary reference point often used in hypervolume calculations. Instead, the origin vector can be used, as all objective ranges are normalized to the $[0, 1]$ interval.

While $HV_{norm}$ addresses scale bias and reference point dependence, it introduces a new challenge: it does not account for variations in the maximally achievable $HV_{norm}$ across contexts. Intuitively, contexts with more convex Pareto fronts allow for higher maximum $HV_{norm}$ values compared to those with concave fronts. Naively aggregating $HV_{norm}$ scores across contexts without accounting for these differences skews evaluations toward contexts with higher achievable hypervolume, unfairly penalizing agents that balance learning across all environments—contradicting the goal of generalization.

To address this, we proposed the NHGR metric, which adjusts for these discrepancies by comparing the generalist agent's performance against the maximum achievable $HV_{norm}$, derived from the optimal Pareto front, in each context. By evaluating performance as a ratio of this maximum, NHGR ensures that contexts with inherently lower achievable hypervolumes are weighted equally against those with higher achievable hypervolumes, enabling fair and unbiased generalization assessments across all contexts.

Table 1 presents the mean performance of the GPI-LS (Alegre et al., 2023) algorithm across three environments in the MO-HalfCheetah domain. As shown, the raw HV metric exhibits performance differences on the order of $10^4$ between environments. This disparity arises because small variations in reward ranges compound in hypervolume calculations, particularly as the number of objectives increases, reducing interpretability. While $HV_{norm}$ mitigates this issue by normalizing objective ranges, it unfairly penalizes the generalist in

|  | Default | Hard | Slippery |
|---|---|---|---|
| HV | $1.4e^5$ | $5.7e^4$ | $9.3e^4$ |
| $HV_{norm}$ | 0.20 | 0.065 | 0.094 |
| NHGR | 0.26 | 0.10 | 0.14 |

Table 1: Illustration of different metrics on 3 MO-HalfCheetah environments.

the *Hard* and *Slippery* environment, where the optimal Pareto front has a significantly lower hypervolume compared to other contexts. In contrast, the NHGR metric offers a more balanced assessment by evaluating the generalist's normalized hypervolume relative to the maximum achievable value. This ensures fair comparisons across all contexts. Additionally, from a generalization perspective, NHGR is more interpretable, as it directly quantifies how close an agent is to achieving the optimal performance in each evaluation context.

| Domains | Metrics | PCN | CAPQL | PGMORL | Envelope | GPI-LS | GPI-PD | MORL/D SB | MORL/D SB+PSA |
|---|---|---|---|---|---|---|---|---|---|
| MO-Lunar Lander | HV ($10^8$) | $6.09 \pm 0.28$ | N/A | N/A | $4.14 \pm 0.65$ | $\mathbf{9.08 \pm 0.55}$ | $7.62 \pm 0.47$ | $8.52 \pm 0.65$ | $8.55 \pm 0.48$ |
|  | EUM ($10^1$) | $-0.45 \pm 0.41$ | N/A | N/A | $-2.53 \pm 0.83$ | $\mathbf{0.94 \pm 0.25}$ | $0.09 \pm 0.30$ | $0.51 \pm 0.29$ | $0.45 \pm 0.21$ |
|  | NHGR | $0.28 \pm 0.04$ | N/A | N/A | $0.18 \pm 0.06$ | $\mathbf{0.66 \pm 0.06}$ | $0.45 \pm 0.07$ | $0.57 \pm 0.07$ | $0.57 \pm 0.05$ |
|  | EUGR | $0.01 \pm 0.03$ | N/A | N/A | $0.00 \pm 0.00$ | $\mathbf{0.50 \pm 0.14}$ | $0.09 \pm 0.10$ | $0.27 \pm 0.13$ | $0.23 \pm 0.10$ |
| MO-Hopper | HV ($10^7$) | $0.44 \pm 0.06$ | $\mathbf{1.40 \pm 0.20}$ | $0.91 \pm 0.18$ | N/A | $1.32 \pm 0.23$ | $1.18 \pm 0.24$ | $1.45 \pm 0.25$ | $\mathbf{1.57 \pm 0.27}$ |
|  | EUM ($10^2$) | $0.59 \pm 0.07$ | $\mathbf{1.45 \pm 0.14}$ | $1.07 \pm 0.14$ | N/A | $\mathbf{1.38 \pm 0.14}$ | $1.26 \pm 0.17$ | $1.45 \pm 0.15$ | $\mathbf{1.50 \pm 0.15}$ |
|  | NHGR | $0.05 \pm 0.03$ | $\mathbf{0.50 \pm 0.16}$ | $0.20 \pm 0.09$ | N/A | $0.45 \pm 0.15$ | $0.37 \pm 0.11$ | $0.46 \pm 0.14$ | $\mathbf{0.54 \pm 0.15}$ |
|  | EUGR | $0.31 \pm 0.05$ | $\mathbf{0.77 \pm 0.12}$ | $0.57 \pm 0.10$ | N/A | $\mathbf{0.74 \pm 0.11}$ | $0.66 \pm 0.10$ | $\mathbf{0.77 \pm 0.12}$ | $\mathbf{0.80 \pm 0.12}$ |
| MO-Half Cheetah | HV ($10^5$) | $0.51 \pm 0.00$ | $0.46 \pm 0.10$ | $0.57 \pm 0.07$ | N/A | $0.81 \pm 0.39$ | N/A | $0.97 \pm 0.32$ | $\mathbf{1.15 \pm 0.36}$ |
|  | EUM ($10^1$) | $0.07 \pm 0.05$ | $-2.86 \pm 5.24$ | $-1.38 \pm 1.46$ | N/A | $0.46 \pm 6.64$ | N/A | $3.28 \pm 2.94$ | $\mathbf{4.27 \pm 3.15}$ |
|  | NHGR | $0.03 \pm 0.02$ | $0.03 \pm 0.03$ | $0.07 \pm 0.02$ | N/A | $0.16 \pm 0.12$ | N/A | $0.21 \pm 0.09$ | $\mathbf{0.29 \pm 0.13}$ |
|  | EUGR | $0.00 \pm 0.00$ | $0.00 \pm 0.00$ | $0.00 \pm 0.01$ | N/A | $\mathbf{0.11 \pm 0.12}$ | N/A | $0.16 \pm 0.12$ | $\mathbf{0.20 \pm 0.13}$ |
| MO-Humanoid | HV ($10^4$) | $3.80 \pm 0.27$ | $\mathbf{5.14 \pm 0.79}$ | $0.15 \pm 0.32$ | N/A | $1.33 \pm 0.09$ | N/A | $4.86 \pm 1.08$ | $4.78 \pm 1.15$ |
|  | EUM ($10^2$) | $0.95 \pm 0.08$ | $\mathbf{1.26 \pm 0.18}$ | $-0.38 \pm 0.25$ | N/A | $0.24 \pm 0.09$ | N/A | $1.27 \pm 0.26$ | $1.24 \pm 0.27$ |
|  | NHGR | $0.24 \pm 0.07$ | $0.31 \pm 0.10$ | $0.03 \pm 0.02$ | N/A | $0.07 \pm 0.04$ | N/A | $\mathbf{0.44 \pm 0.19}$ | $0.43 \pm 0.20$ |
|  | EUGR | $0.42 \pm 0.09$ | $\mathbf{0.55 \pm 0.13}$ | $0.01 \pm 0.01$ | N/A | $0.10 \pm 0.04$ | N/A | $\mathbf{0.56 \pm 0.17}$ | $\mathbf{0.55 \pm 0.18}$ |
| MO-Super MarioBros | HV ($10^6$) | $\mathbf{1.41 \pm 0.27}$ | N/A | N/A | $\mathbf{1.48 \pm 0.24}$ | $\mathbf{1.45 \pm 0.19}$ | N/A | N/A | N/A |
|  | EUM ($10^1$) | $\mathbf{1.22 \pm 0.81}$ | N/A | N/A | $\mathbf{1.41 \pm 0.70}$ | $\mathbf{1.30 \pm 0.52}$ | N/A | N/A | N/A |
|  | NHGR | $0.04 \pm 0.06$ | N/A | N/A | $\mathbf{0.08 \pm 0.08}$ | $\mathbf{0.12 \pm 0.14}$ | N/A | N/A | N/A |
|  | EUGR | $\mathbf{0.40 \pm 0.16}$ | N/A | N/A | $\mathbf{0.47 \pm 0.14}$ | $\mathbf{0.47 \pm 0.19}$ | N/A | N/A | N/A |
| MO-LavaGrid | HV ($10^5$) | $1.12 \pm 0.67$ | N/A | N/A | $1.10 \pm 0.57$ | $1.12 \pm 0.58$ | N/A | $\mathbf{3.54 \pm 1.88}$ | $2.76 \pm 1.84$ |
|  | EUM ($10^2$) | $-2.06 \pm 0.61$ | N/A | N/A | $-1.95 \pm 0.48$ | $-1.97 \pm 0.51$ | N/A | $\mathbf{-0.85 \pm 0.95}$ | $-1.34 \pm 1.05$ |
|  | NHGR | $0.03 \pm 0.06$ | N/A | N/A | $0.02 \pm 0.05$ | $0.02 \pm 0.05$ | N/A | $\mathbf{0.37 \pm 0.21}$ | $0.27 \pm 0.23$ |
|  | EUGR | $0.00 \pm 0.00$ | N/A | N/A | $0.00 \pm 0.00$ | $0.00 \pm 0.00$ | N/A | $\mathbf{0.06 \pm 0.15}$ | $0.04 \pm 0.13$ |

Table 2: Performance of various MORL algorithms across different domains and evaluation metrics. Higher values indicate better performance for all metrics. Each entry indicates the mean and standard deviation computed over 5 independent runs. Bolded values fall within one standard deviation of the best mean.

## D.2 EXPECTED UTILITY GENERALIZATION RATIO

When a good prior over possible user utility functions is known, The *Expected Utility Metric* (EUM) proposed by Zintgraf et al. (2015) can be used for evaluating MORL algorithms. This metric calculates the expected utility of the agent's approximate Pareto front under the prior distribution of user utility functions, parameterized by a weight space $\mathcal{W}$. A higher EUM indicates that the policies yields better expected utility across the user utility functions. Under the SER criterion, the expected utility of the approximate Pareto front $\tilde{\mathcal{F}}$ produced by a MORL agent is given by:

$$\text{EUM}(\tilde{\mathcal{F}}) = \mathbb{E}_{\mathbf{w} \sim \mathcal{W}} \left[ \max_{\mathbf{v}^\pi \in \tilde{\mathcal{F}}} u(\mathbf{v}^\pi, \mathbf{w}) \right],$$

where $u$ is the user's utility function, and $\mathbf{v}^\pi$ is the expected value vector of policy $\pi$ within the Pareto set.

There are several scenarions in which the EUM can be applied. In practical applications, the utility function of the stakeholders might be known due to domain knowledge. Using EUM would therefore allow for more direct evaluations on how the solutions generated by the MORL agent corresponds to improving the utility of the stakeholders. Not every point on the Pareto front would contribute to an increase in the EUM for a given utility function. In addition, the hypervolume metric is known for its computational challenge especially in higher dimensions, although various approximation algorithms and heuristics have been developed to estimate the hypervolume more efficiently. The EUM, on the other hand, depends only on the number of solutions on the approximate Pareto front and the number of weights sampled from the weight space.

As mentioned in Section 4 of the main body, when aggregating performances across multiple contexts for measuring generalization, we must ensure that each context is equally attributed. Specifically, we can calculate a variant on the NHGR metric we call the *Expected Utility Generalization Ratio* (EUGR).

**Definition 5** (Expected Utility Generalization Ratio). Let $\tilde{\mathcal{F}}_c$ and $\mathcal{F}_c^*$ be the approximate Pareto front obtained by generalist MORL agent and the optimal Pareto front for context $c$ respectively. The EUGR for the agent in $c$ is defined as:

$$\text{EUGR}(\tilde{\mathcal{F}}_c, \mathcal{F}_c^*) = \frac{\text{EUM}(\tilde{\mathcal{F}}_c)}{\text{EUM}(\mathcal{F}_c^*)}.$$

Unlike in NHGR, the Pareto front is not normalized here. This is because the utility functions used in the EUM should already inherently reflect the stakeholders' preferences over objectives, including their relative importance.

Since there is no clear utility function distribution that is suited for toy domains like those in our benchmark, we followed standard conventions and evaluated EUM and EUGR in Table 2 using linear utility functions with weights summing to unity:

$$u(\mathbf{v}^\pi, \mathbf{w}) = \mathbf{w}^\mathsf{T} \mathbf{v}^\pi \quad \text{where} \quad \sum_i w_i = 1, \ w_i \geq 0, \ i = 1, \ldots, k$$

where $k$ is the number of objectives. Specifically, we scalarized the value vectors in the approximate Pareto front obtained by each algorithm at the end of each run using 100 weight vectors sampled uniformly from the unit weight simplex to compute the expected utility.

## D.3 SINGLE OBJECTIVE UTILITY FUNCTION RESULTS

As discussed in Section 6.2, most classic SORL domains in Gymnasium inherently perform an implicit scalarization of multiple objectives when defining the scalar reward function. As such, for every domain, we track and plot the single-objective scalarization function $f_{\text{SORL}}$ returns for all MORL algorithms, as well as for the SAC algorithm, across all evaluation episodes throughout training. Across many domains in our benchmark, we observe that the leading MORL algorithms could outperform or achieve comparable performance to the single-objective SAC algorithm in terms of maximum $f_{\text{SORL}}$ return. Below, we present the $f_{\text{SORL}}$ equations for each environment along with their corresponding plots.

### D.3.1 MO-HOPPER

The default single objective utility function of the MO-Hopper domain is same as the one used in Gymnasium's Hopper, which is

$$f_{\text{SORL}} = 1.5v_x + 0.001c + h$$

where $v_x$ is the forward speed, $c$ is the control cost and $h$ is the reward for staying alive.

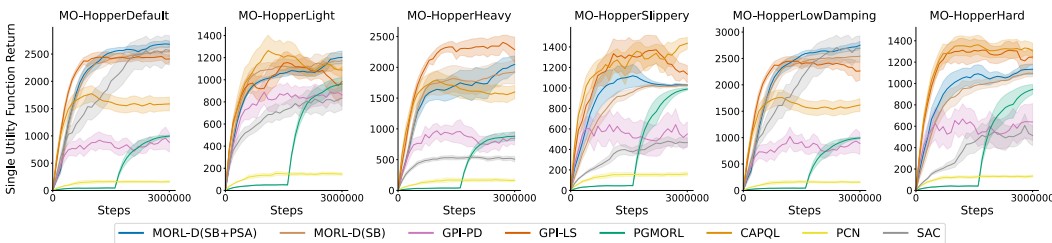

Figure 7: Single-objective return on 6 MO-Hopper testing environments during training. Each curve is measured across 5 seeds (mean and standard error).

### D.4 MO-HALFCHEETAH

The default single objective utility function of the MO-HalfCheetah domain is same as the one used in Gymnasium's HalfCheetah, which is

$$f_{\text{SORL}} = 1.0v_x + 0.1c$$

where $v_x$ is the forward reward and $c$ is the control cost. The HalfCheetah is always alive so it has no alive reward.

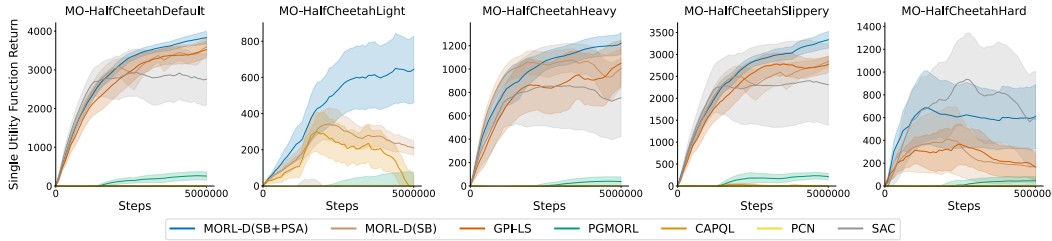

Figure 8: Single-objective return on 5 MO-HalfCheetah testing environments during training. Each curve is measured across 5 seeds (mean and standard error).

### D.4.1 MO-HUMANOID

The single objective utility function of the MO-Humanoid domain is

$$f_{\text{SORL}} = 1.25v_x + 0.001c + 2.0h$$

where $v_x$ is the forward speed, $c$ is the control cost and $h$ is the reward for staying alive. The original Gymansium's Humanoid domain uses a 5.0 coefficient for the alive reward but we tuned it down to because it dominating all the other objectives in terms of magnitude. We have also verified that the convergence of the SAC agent on the original single-objective Humanoid environment remains unchanged with this lower alive reward.

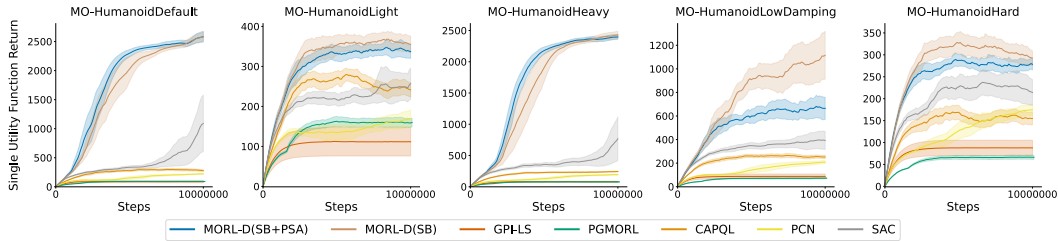

Figure 9: Single-objective return on 5 MO-Humanoid testing environments during training. Each curve is measured across 5 seeds (mean and standard error).

### D.4.2 MO-LUNARLANDER

The default single objective utility function of the MO-LunarLander domain is same as the one used in Gymnasium's LunarLander, which is

$$f_{\text{SORL}} = l + s + 0.3mc + 0.03sc$$

where $l$ is a -100/+100 one-time reward if the lander lands successfully or crashes, $s$ is the shaping reward, $mc$ is the main engine cost and $sc$ is the side engine cost.

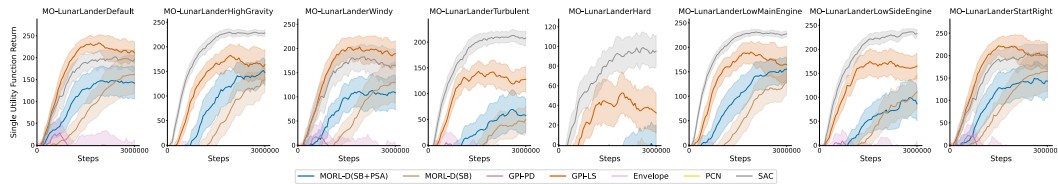

Figure 10: Single-objective return on 8 MO-LunarLander testing environments during training. Each curve is measured across 5 seeds (mean and standard error).

### D.4.3 MO-SUPERMARIOBROS

The default single objective utility function of the MO-SuperMarioBros domain is same as the one used in Gym Super Mario Bros (Kauten, 2018), which is

$$f_{\text{SORL}} = f + t + d$$

where $f$ is a forward reward, $t$ is the time penalty, $d$ is the death penalty.

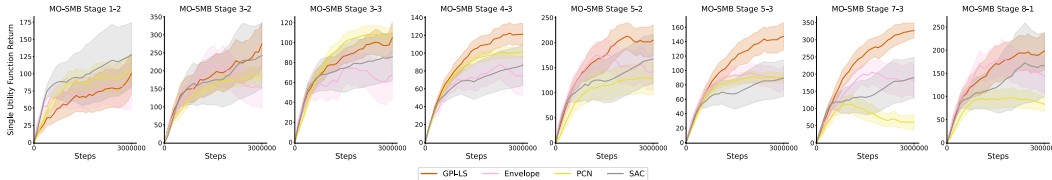

Figure 11: Single-objective return on 5 MO-SuperMarioBros (abbreviated as MO-SMB) testing environments during training. Each curve is measured across 5 seeds (mean and standard error).

## E  TRAINING DETAILS

Table 3 shows shared training hyperparameters across algorithms for each domain in the MORL generalization benchmark. The scripts to reproduce the results in this paper are provided in the codebase, alongside with more specific hyperparameters for the different algorithms. To have fair evaluations, we utilize the same architectures for the policy and value functions across all algorithms for each domain. Specifically, for MO-LavaGrid, MO-LunarLander, MO-Hopper, MO-HalfCheetah,

and MO-Humanoid, the policy and value functions are multi-layer perceptrons (MLPs) with four hidden layers of 256 units each. For MO-SuperMarioBros which has image observations, the policy and value functions consist of a NatureCNN (Mnih et al., 2015) followed by a MLP with two hidden layers of 512 units each. For off-policy algorithms that depend on experience replay, we ensure the same replay buffer size is used. We direct researchers to our codebase for detailed scripts and hyperparameter settings used in training each baseline algorithm for the evaluations in Section 6.

| Parameter | MO-LavaGrid | MO-Lunar Lander | MO-Super MarioBros | MO-Hopper | MO-Half Cheetah | MO-Humanoid |
|---|---|---|---|---|---|---|
| Discount $\gamma$ | 0.995 | 0.99 | 0.99 | 0.99 | 0.99 | 0.99 |
| Adam learning rate | $3e^{-4}$ | $3e^{-4}$ | $3e^{-4}$ | $3e^{-4}$ | $3e^{-4}$ | $3e^{-4}$ |
| Adam $\epsilon$ | $1e^{-8}$ | $1e^{-8}$ | $1e^{-8}$ | $1e^{-8}$ | $1e^{-8}$ | $1e^{-8}$ |
| Batch Size | 128 | 128 | 64 | 256 | 256 | 256 |
| Replay buffer size | $1e^6$ | $1e^6$ | $1e^5$ | $1e^6$ | $1e^6$ | $1e^6$ |
| Max episode steps | 256 | 1000 | 2000 | 1000 | 1000 | 1000 |
| Env Steps | $5e^6$ | $3e^6$ | $3e^6$ | $3e^6$ | $5e^6$ | $1e^7$ |

Table 3: Hyperparameters used for training on MORL generalization benchmark.

## F    MORL-GENERALIZATION BENCHMARK DETAILS

In this section, we begin by providing an overview of the domains in our *MORL-Generalization* benchmark, highlighting the distinct context variations each domain introduces. Next, for each domain, we detail the environment parameters used to create their evaluation environments. The code commands to initialize these environments using Gymnasium are also included within our codebase.

### F.1    BENCHMARK DOMAIN DETAILS

Kirk et al. (2023) identified four key types of domain variations for studying generalization: 1) state-space variation (S), which alters the initial state distribution, 2) dynamics variation (D), which alters the transition function, 3) visual variation (O), which impacts the observation function, and 4) reward function variation (R). We provide detailed descriptions of the benchmark domains introduced in Section 5 below:

**MO-LunarLander** (D+S) This is a multi-objective adaptation of Gymnasium's *LunarLander* domain where the agent has to balance between successfully landing on the moon surface, the stability of the spacecraft, the fuel cost of the main engine, and the fuel cost of the side engine. The agent operates over discrete-action and continuous-observation spaces. We introduce a 7-dimensional parameter that varies the environment's gravity, wind power, turbulence, and the lander's main engine power, side engine power, and initial x, y coordinates.

**MO-Hopper** (D) This is a multi-objective adaptation of Gymnasium's *Hopper* domain. The one-legged agent must balance optimizing for its forward velocity, torso height, and energy cost. The agent operates over continuous action and observation spaces. We introduce an 8-dimensional parameter that varies the hopper's body masses (4D), joint damping (3D), and the floor's friction.

**MO-HalfCheetah** (D) This is a multi-objective adaptation of Gymnasium's *HalfCheetah* domain. The 2-dimensional cheetah robot must balance optimizing for its forward velocity and energy cost. The agent operates over continuous action and observation spaces. We introduce an 8-dimensional parameter that varies the cheetah's body masses (7D) and the floor's friction.

**MO-Humanoid** (D) This is a multi-objective adaptation of Gymnasium's *Humanoid* domain. The humanoid robot must balance between optimizing for its forward velocity and its energy cost. The agent operates over continuous action and observation spaces. We introduce a 30-dimensional environment parameter that varies the humanoid's body masses (13D) and joint damping (17D).

**MO-SuperMarioBros** (S) This is a multi-objective adaptation of the *Gym Super Mario Bros* (Kauten, 2018) domain based on the popular Super Mario Bros video game. The agent has to balance between moving forward, collecting coins, and increasing the game score (by stomping enemies, breaking bricks, etc.). The agent operates over discrete-action and discrete-observation (pixel images) spaces. We introduce a 2-dimensional parameter that controls which stage of the Super Mario Bros game to place the agent in. There are a total of 32 possible stages.

**MO-LavaGrid** (S+R) This is a novel multi-objective domain based on *MiniGrid* (Chevalier-Boisvert et al., 2023). The agent (red triangle) has to navigate a 11 x 11 grid, incurring a penalty each time it touches lava and another for every step it takes to collect all 3 goals (blue, green, and yellow blocks), after which the episode terminates. The placements of the agent, goals, and lava blocks are fully configurable, providing diverse evaluation contexts. Additionally, we introduce a 3-dimensional parameter controlling the reward weight of each goal. These weights are concatenated with the state space, ensuring the agent has the necessary information about the reward function to plan its trajectory while balancing goal collection and lava avoidance. The agent operates over discrete-action and mixed continuous-discrete (because of the reward weights) observation spaces.

## F.2 MO-LAVAGRID

The environment parameters for the MO-LavaGrid domain are represented using bit maps, which we are unable to directly translate into this paper. Instead, the evaluation environments are visually shown in Fig. 12. Also, as mentioned in 5, the MO-LavaGrid environment has a 3-dimensional parameter controlling the reward weightages of each goal square (green, blue, yellow). The reward weights for each goal are concatenated to the state space of the agent, and the weights sum to unity. The reward weightages for each goal in each evaluation environment are shown in Table 4.

During training using domain randomization, after each episode concludes, the agent's start position and orientation, the number of lava blocks, the placement of the goals and lava blocks, and reward weightages of the goals are all randomly set. When an agent has collected/visited a goal, the weightage of the goal in the state space is set to 0, to indicate that the reward corresponding to that goal has already been awarded.

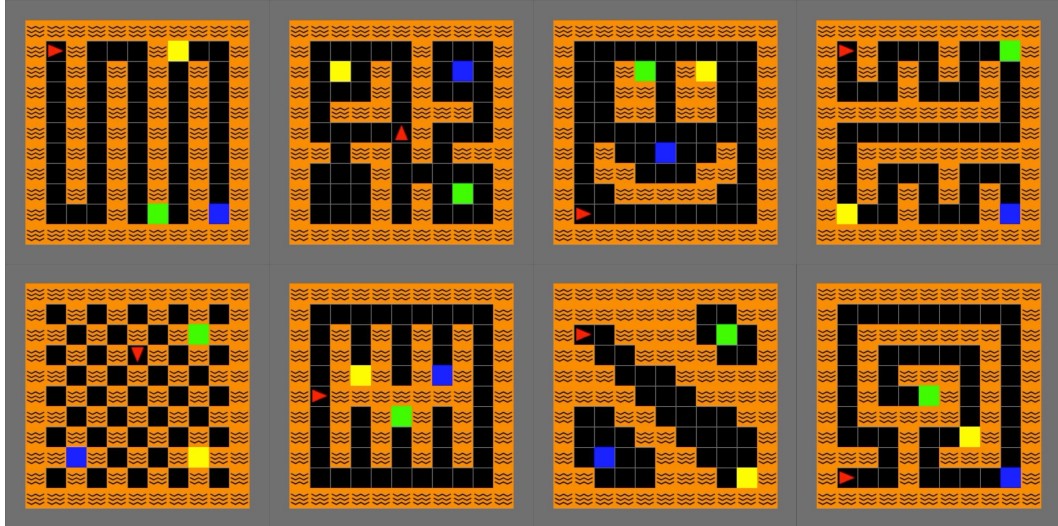

Figure 12: MO-LavaGrid Evaluation Environments. Top row (left to right): MO-LavaGridSnake, MO-LavaGridRoom, MO-LavaGridSmiley, MO-LavaGridMaze. Bottom row (left to right): MO-LavaGridCheckerBoard, MO-LavaGridCorridor, MO-LavaGridIslands, MO-LavaGridLabyrinth

| Environment | Green | Yellow | Blue |
|---|---|---|---|
| MO-LavaGridSnake | 0.20 | 0.30 | 0.50 |
| MO-LavaGridRoom | 0.50 | 0.30 | 0.20 |
| MO-LavaGridSmiley | 0.40 | 0.40 | 0.20 |
| MO-LavaGridMaze | 0.05 | 0.05 | 0.90 |
| MO-LavaGridCheckerBoard | 0.30 | 0.10 | 0.60 |
| MO-LavaGridCorridor | 0.60 | 0.10 | 0.30 |
| MO-LavaGridIslands | 0.3$\dot{3}$ | 0.3$\dot{3}$ | 0.3$\dot{3}$ |
| MO-LavaGridLabyrinth | 0.50 | 0.05 | 0.45 |

Table 4: Reward weightages for MO-LavaGrid evaluation environments.

### F.3 MO-SUPERMARIOBROS

In MO-SuperMarioBros, each environment configuration is instantiated via a 2-dimensional parameter. The first dimension has discrete values $\{1, 2, 3, 4, 5, 6, 7, 8\}$, and indicates the SuperMarioBros world. The second dimension has discrete values $\{1, 2, 3, 4\}$, and indicates the level within the chosen world. Together, the parameters `<world>-<level>` defines the stage (configuration) of the environment.

During training using domain randomization, an environment is randomly selected from the 32 possible stages, except Stage 3-3 which is reserved for zero-shot generalization evaluation. During evaluation, the agents are evaluated on only 8/32 stages to keep the runtime within reasonable limits. The evaluation stages are visually shown in Fig. 13. The evaluation stages are carefully selected to encompass a wide range of environment dynamics and visual renditions. Additionally, they are chosen to ensure that each stage offers non-zero rewards across all objective dimensions. This is crucial to prevent hypervolume evaluations from collapsing to zero, which would occur if any dimension of the objective space had a zero achievable range.

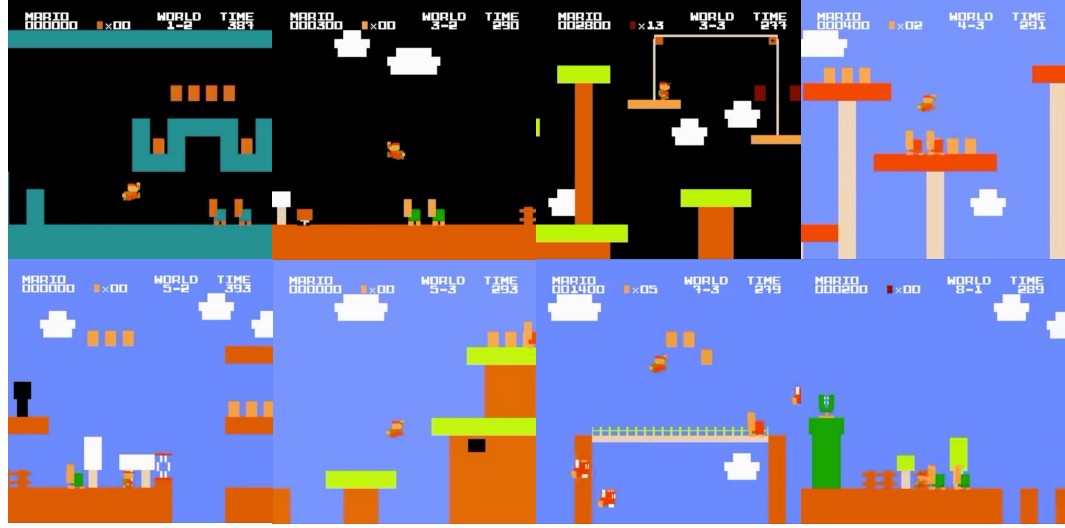

Figure 13: MO-SuperMarioBros Evaluation Environments. Top row (left to right): Stage 1-2, Stage 3-2, Stage 3-3 (zero shot), Stage 4-3. Bottom row (left to right): Stage 5-2, Stage 5-3, Stage 7-3, Stage 8-1

### F.4 MO-LUNARLANDER

In MO-LunarLander, each environment configuration is instantiated via a 7-dimensional parameter. The dimensions of the environment parameter corresponds to the gravity coefficient, wind power, turbulence, the lander's main engine power, the lander's side engine power, the lander's initial x-coordinate, and the lander's initial y-coordinate.

During evaluation, we assess the agents performances on a predefined set of 8 environment configurations: Default, High Gravity, Windy, Turbulent, Low Main Engine, Low Side Engine, Start Right, and Hard. Table 5 displays the environment parameter values used for each environment configuration.

| Parameters | Default | High Gravity | Windy | Turbulent | Low Main Engine | Low Side Engine | Start Right | Hard |
|---|---|---|---|---|---|---|---|---|
| Gravity | -10.0 | -13.0 | -10.0 | -10.0 | -10.0 | -10.0 | -10.0 | -12.0 |
| Wind Power | 15.0 | 15.0 | 20.0 | 15.0 | 15.0 | 15.0 | 15.0 | 17.0 |
| Turbulence Power | 1.5 | 1.5 | 1.5 | 3.5 | 1.5 | 1.5 | 1.5 | 2.5 |
| Main Engine Power | 13.0 | 13.0 | 13.0 | 13.0 | 10.0 | 13.0 | 13.0 | 12.0 |
| Side Engine Power | 0.6 | 0.6 | 0.6 | 0.6 | 0.6 | 0.3 | 0.6 | 0.4 |
| Initial X Coeff | 0.5 | 0.5 | 0.5 | 0.5 | 0.5 | 0.4 | 0.75 | 0.4 |
| Initial Y Coeff | 1.0 | 1.0 | 1.0 | 1.0 | 1.0 | 1.0 | 1.0 | 1.0 |

Table 5: Environment parameters for MO-LunarLander

## F.5 MO-HOPPER

In MO-Hopper, each environment configuration are instantiated via a 8-dimensional parameter that varies the hopper's body masses (4D), joint damping (3D), and the floor's friction (1D).

During evaluation, we assess the agents performances on a predefined set of 6 environment configurations: Default, Light, Heavy, Slippery, Low Damping, and Hard. Table 6 displays the environment parameter values used for each environment configuration.

| Parameters | Default | Light | Heavy | Slippery | Low Damping | Hard |
|---|---|---|---|---|---|---|
| Torso Mass | 3.7 | 0.5 | 9.0 | 3.7 | 3.7 | 0.1 |
| Thigh Mass | 4.0 | 0.5 | 9.0 | 4.0 | 4.0 | 9.0 |
| Leg Mass | 2.8 | 0.3 | 8.5 | 2.8 | 2.8 | 9.0 |
| Foot Mass | 5.3 | 0.7 | 10.0 | 5.3 | 5.3 | 0.1 |
| Damping 0 | 1.0 | 1.0 | 1.0 | 1.0 | 0.1 | 0.1 |
| Damping 1 | 1.0 | 1.0 | 1.0 | 1.0 | 0.1 | 0.1 |
| Friction | 1.0 | 1.0 | 1.0 | 0.1 | 1.0 | 0.1 |

Table 6: Environment parameters for MO-Hopper

## F.6 MO-HALFCHEETAH

In MO-HalfCheetah, each environment configuration is instantiated via a 8-dimensional parameter that varies the cheetah's body masses (7D) and the floor's friction (1D).

During evaluation, we assess the agents performances on a predefined set of 5 environment configurations: Default, Light, Heavy, Slippery, and Hard. Table 7 displays the environment parameter values used for each environment configuration.

| Parameters | Default | Light | Heavy | Slippery | Hard |
|---|---|---|---|---|---|
| Torso Mass | 6.25 | 0.5 | 10.0 | 6.25 | 6.25 |
| Back Thigh Mass | 1.538 | 0.1 | 9.5 | 1.54 | 9.5 |
| Back Shin Mass | 1.441 | 0.1 | 9.5 | 1.59 | 9.5 |
| Back Foot Mass | 0.891 | 0.1 | 9.5 | 1.10 | 9.5 |
| Front Thigh Mass | 1.434 | 0.1 | 9.5 | 1.44 | 0.1 |
| Front Shin Mass | 1.198 | 0.1 | 9.5 | 1.20 | 0.1 |
| Front Foot Mass | 0.869 | 0.1 | 9.5 | 0.88 | 0.1 |
| Friction | 0.4 | 0.4 | 0.4 | 0.1 | 0.1 |

Table 7: Environment parameters for MO-HalfCheetah

## F.7 MO-HUMANOID

In MO-Humanoid, each environment configuration is instantiated via a 30-dimensional parameter that varies the humanoid's body masses (13D) and joint damping (17D).

During evaluation, we assess the agents performances on a predefined set of 5 environment configurations: Default, Light, Heavy, Low Damping, and Hard. Table 8 displays the environment parameter values used for each environment configuration.

| Parameters | Default | Light | Heavy | Low Damping | Hard |
|---|---|---|---|---|---|
| Mass 1 | 8.91 | 1.7 | 10.0 | 8.91 | 8.91 |
| Mass 2 | 2.26 | 0.5 | 7.0 | 2.26 | 2.26 |
| Mass 3 | 6.62 | 1.3 | 9.0 | 6.62 | 6.62 |
| Mass 4 | 4.75 | 0.7 | 8.0 | 4.75 | 0.7 |
| Mass 5 | 2.76 | 0.6 | 7.0 | 2.76 | 0.6 |
| Mass 6 | 1.77 | 0.5 | 6.0 | 1.77 | 0.5 |
| Mass 7 | 4.75 | 0.7 | 8.0 | 4.75 | 8.0 |
| Mass 8 | 2.76 | 0.5 | 7.0 | 2.76 | 7.0 |
| Mass 9 | 1.77 | 0.3 | 6.0 | 1.77 | 6.0 |
| Mass 10 | 1.66 | 0.3 | 6.0 | 1.66 | 0.1 |
| Mass 11 | 1.23 | 0.1 | 5.5 | 1.23 | 0.1 |
| Mass 12 | 1.66 | 0.3 | 6.0 | 1.66 | 5.0 |
| Mass 13 | 1.23 | 0.1 | 5.5 | 1.23 | 5.0 |
| Damp 1 | 1.0 | 5.0 | 5.0 | 1.0 | 1.0 |
| Damp 2 | 1.0 | 5.0 | 5.0 | 1.0 | 1.0 |
| Damp 3 | 1.0 | 5.0 | 5.0 | 1.0 | 1.0 |
| Damp 4 | 1.0 | 5.0 | 5.0 | 1.0 | 1.0 |
| Damp 5 | 1.0 | 5.0 | 5.0 | 1.0 | 1.0 |
| Damp 6 | 1.0 | 5.0 | 5.0 | 1.0 | 1.0 |
| Damp 7 | 0.2 | 1.0 | 1.0 | 0.2 | 0.2 |
| Damp 8 | 1.0 | 5.0 | 5.0 | 1.0 | 1.0 |
| Damp 9 | 1.0 | 5.0 | 5.0 | 1.0 | 1.0 |
| Damp 10 | 1.0 | 5.0 | 5.0 | 1.0 | 1.0 |
| Damp 11 | 0.2 | 1.0 | 1.0 | 0.2 | 0.2 |
| Damp 12 | 0.2 | 1.0 | 1.0 | 0.2 | 0.2 |
| Damp 13 | 0.2 | 1.0 | 1.0 | 0.2 | 0.2 |
| Damp 14 | 0.2 | 1.0 | 1.0 | 0.2 | 0.2 |
| Damp 15 | 0.2 | 1.0 | 1.0 | 0.2 | 0.2 |
| Damp 16 | 0.2 | 1.0 | 1.0 | 0.2 | 0.2 |
| Damp 17 | 0.2 | 1.0 | 1.0 | 0.2 | 0.2 |

Table 8: Environment parameters for MO-Humanoid

