# OpenReview forum: "On Generalization Across Environments In Multi-Objective Reinforcement Learning"
_ICLR.cc/2025/Conference — ICLR 2025 Poster_

### Official Review · Reviewer_nVsL · 2024-10-22

**Soundness:** 2
**Presentation:** 2
**Contribution:** 2
**Rating:** 6
**Confidence:** 4

**Summary:**

This paper introduces a framework and testbed for evaluating generalization in multi-objective reinforcement learning (MORL). **The authors claim that generalization in MORL is a crucial research direction, but existing MORL algorithms are not sufficient for MORL generalization.** To support the claim, the authors propose the concept of a multi-objective contextual MDP and introduce the normalized hypervolume generalization ratio (NHGR) as a metric for assessing generalization in MORL. To empirically test this, they modify existing MORL environments by adjusting key environment parameters and applying domain randomization techniques during training. The paper reports NHGR performance across different MORL algorithms during the evaluation phase and concludes by highlighting the need for further research on MORL generalization.

**Strengths:**

* Considering generalization in MORL is a sound direction in the RL community.
* The necessity of MORL generalization is empirically demonstrated in Section 6.2, partly supporting the authors' claim.

**Weaknesses:**

1. **The overall contribution and novelty of this paper remain incomplete.** While the paper highlights a promising research direction, it lacks a concrete solution for addressing the problem of MORL generalization. To enhance the paper’s completeness, the authors should propose and integrate their own method to tackle MORL generalization effectively.

2. **The authors' claim is not fully supported due to the unclear soundness of the proposed NHGR metric.**
* 2-1. For clarity, does the combined specialist Pareto front consist of the *filtered* nondominated value vectors from the *union* of nondominated value vectors across the 8 MORL algorithms, each trained individually on a specific context?

* 2-2. The computation of NHGR is highly demanding, especially as the number of MORL algorithms and contexts increases during evaluation. If performance can be directly computed for any given context, there would be little need for a generalized algorithm, as the "combined specialist" would suffice. This raises concerns about NHGR's practical applicability in scenarios with large computational requirements.

* 2-3. A key weakness of NHGR is that its denominator relies on the performance of existing MORL algorithms, which may not perform well in certain contexts or in environments with many objectives. In such cases, where the "combined specialist" performs poorly, NHGR becomes an unreliable metric.

* 2-4. The necessity for normalization in NHGR is questionable, especially since true upper and lower bounds for each objective are not typically known a priori. If these bounds are not tightly set, the reliability of NHGR results is compromised. Moreover, determining accurate bounds (e.g., $v^c_{min}$ and $v^c_{max}$) requires running multiple MORL algorithms, increasing computational complexity. Establishing these bounds is itself a challenging task, particularly during online training.


* 2-5. A potentially better approach would be to use the expected standard hypervolume across evaluation contexts, supplemented by the expected standard sparsity to address the limitations of relying solely on hypervolume.


* 2-6. Figure 1 does not align with the definition of normalization bounds $v^c_{min}$ and $v^c_{max}$. The reference point is determined by the x-coordinate of the upper-leftmost point and the y-coordinate of the lower-rightmost point.


* 2-7. For clarity, is the number of the episodes evaluated for each context is set at 100/(number of evaluation contexts) (excluding Mario)?


3. **The paper’s organization needs improvement.**
- 3-1.  It would improve the flow of the paper if Section 6.2 were moved closer to the background section to better emphasize the motivation behind the work.
- 3-2. Including the related work section in the main body of the paper would enhance readability, as it is critical for contextual understanding.
- 3-3. A more detailed explanation of how each MORL algorithm operates, such as the weight adaptation variant of MORL/D SB, would be beneficial (this could be placed in the Appendix for conciseness).

4. **Several technical clarifications are needed.**
- 4-1. Is SAC trained with a standard scalar reward function (described in Appendix D) for each context vector independently, or does it follow domain randomization (i.e., sampling random environment parameters for each episode)? I assume the authors intended the latter.
- 4-2. In Line 479, the phrase "...scalarized them using $f_{SORL}$​..." is unclear. Additionally, how many solution vectors are sampled for each algorithm?
- 4-3. How is the optimality gap calculated?
- 4-4. For the SAC implementation in Mario and LavaGrid, is it "SAC-discrete"? If so, how was SAC-discrete implemented, and how does its performance compare to DQN?
- 4-5. In Section 6.2, what is meant by the "upward speed" in the scalar reward formulation (Line 461 and Figure 6)?

5. Minor Suggestions:
- I recommend dividing paragraphs to improve readability.
- Line 302: "We introduce an 8-dimensional parameter..." should be corrected to "7-dimensional."

**Questions:**

Please see the weaknesses part above.

**Post Rebuttal Comment**


After careful consideration, I have revised my final rating.

As the authors acknowledged, the proposed metric has inherent limitations and these can be mitigated by also providing standard metrics. Each standard metric in MORL - hypervolume, expected utility metric, sparsity, and so on - has its strengths and weaknesses; therefore, the best approach can be to report multiple metrics to complement one another. This principle can be applied to the paper's setting as well. Introducing a new evaluation metric for MORL is a promising research direction, and it may be harsh to penalize the effort solely because the metric is not perfect.

I hope the authors include a thorough discussion on the metric's limitations and potential mitigations in a future version, if possible.

---

> ### Author Response · Authors · 2024-11-23
> **Response to Reviewer nVsL (Part 1/3)**
>
> We sincerely thank the reviewer for taking the time to write such a detailed review amongst the many other papers assigned for reviewing during this conference. The following clarifications will explain our stance and strengthen the case for our paper. We kindly request the reviewer's time to look through them. For all revisions to the manuscript, they are highlighted in $\color{teal}{\text{teal}}$.
>
> ---
>
> ## Clarifying questions
>
> First, we hope to get the clarification questions out of the way before responding to the other points regarding weaknesses of the paper.
>
> > 2-1. For clarity, does the combined specialist Pareto front consist of the filtered nondominated value vectors from the union of nondominated value vectors across the 8 MORL algorithms, each trained individually on a specific context?
>
> Yes, that is the case.
>
> > 2-6 Figure 1 does not align with the definition of normalization bounds $v_{min}^c$ and $v_{max}^c$. The reference point is determined by the x-coordinate of the upper-leftmost point and the y-coordinate of the lower-rightmost point.
>
> Appreciate the sharp eye, fixed!
>
> > 2-7. For clarity, is the number of the episodes evaluated for each context is set at 100/(number of evaluation contexts) (excluding Mario)?
>
> The number of episodes evaluated is **100 $\times$ (number of evaluation contexts)**, except Mario (replace 100 with 32).
>
> > 3-1 to 3-3 Organization suggestions
>
> We sincerely thank the reviewer for your attention to detail and suggestions. We agree that your suggestions would make the paper much stronger. 3-2 has been completed. 3-3 has been included in Appendix E and F. 3-1 might become problematic for the flow of the paper because Section 6.2 depends on Sections 5 and 6 (introduction of experiments and benchmarks). But we will see how we can make it work!
>
> > 4-1. Is SAC trained with a standard scalar reward function (described in Appendix D) for each context vector independently, or does it follow domain randomization (i.e., sampling random environment parameters for each episode)? I assume the authors intended the latter.
>
> Yep, it is the latter.
>
> > In Line 479, the phrase "...scalarized them using $f_{\text{SORL}}$..." is unclear. Additionally, how many solution vectors are sampled for each algorithm?
>
> The number of solution vectors sampled is same as the experiments in Section 6.1, i.e. 100 solution vectors for all domains except 32 in MO-SuperMarioBros. Scalarizing using $f_{\text{SORL}}$ refers to using the default single-objective reward function in Gymnasium. This is because many Gymnasium environments are implicitly multi-objective. For example, the single-objective RL agent for Hopper is trained with the reward function $f_{\text{SORL}} = 1.5 v_x + 0.001c + h$, where each component corresponds to forwards velocity, control cost and alive bonus respectively. This is a reward function probably engineered by the original Mujoco/OpenAI Gym developers via many trial-and-errors to get the SORL agent to move forward. Such **implicit scalarization and inherent multi-objectivity** exists within all other single-objective Gymnasium environments as well, a less-known fact to most developers who invoke the environment via Gymnasium API.
>
> > 4-3. How is the optimality gap calculated?
>
> Optimality gap corresponds to the area above the performance curve below an optimal threshold line (in our case, NHGR = 1). Together with IQM, these methods of aggregating performances across many test contexts helps to counter the inherent statistical uncertainty in deep RL. These metrics are adapted from the library provided by Agarwal et al. (2021) and more information can be found in that paper.
>
> [1] Rishabh Agarwal, Max Schwarzer, Pablo Samuel Castro, Aaron C Courville, and Marc Bellemare. Deep reinforcement learning at the edge of the statistical precipice. Advances in neural informa- tion processing systems, 34:29304–29320, 2021.
>
> > 4-4. For the SAC implementation in Mario and LavaGrid, is it "SAC-discrete"?
>
> We primarily focussed on SAC-discrete since MORL/D (SB) and MORL/D (SB+PSA) both use the same CleanRL SAC-discrete backboneand both algorithms achieved leading performances in MO-Lavagrid and MO-LunarLander.
>
> > 4-5. In Section 6.2, what is meant by the "upward speed" in the scalar reward formulation?
>
> It is calculated using the change in distance in the y-axis divided by number of frames between timesteps, similar to how the main reward (forward speed/velocity) is calculated.  On hindsight, "upwards velocity" would have been a clearer term, thank you for bringing this up.
>
> > Line 302: "We introduce an 8-dimensional parameter..." should be corrected to "7-dimensional."
>
> The MO-Hopper environment has parameters: body masses (4D), joint damping (3D), and floor friction (1D).

---

> ### Author Response · Authors · 2024-11-23
> **Response to Reviewer nVsL (Part 2/3)**
>
> ## Strengthening the case for NHGR
>
> >  2-2. If performance can be directly computed for any given context, there would be little need for a generalized algorithm, as the "combined specialist" would suffice.
>
> We believe there may have been a misunderstanding here and appreciate the opportunity to clarify. Generalization evaluations in toy environments/simulations serve as controlled settings to systematically compare the different algorithms proposed by researchers. As such, NHGR provides a basis for fair evaluation of generalizability in MORL algorithms in a **benchmark setting**. The combined specialist is only calculated for a set of evaluation contexts, and does not encompass the whole context space which the agent can be deployed in. Therefore, the combined specialist cannot be used in place of the generalist agent (which seeks to generalise across the entire context space). The next point will further emphasize the motivations and importance of NHGR for proper evaluations of MORL generalization.
>
> > 2-3 + 2-4. A key weakness of NHGR is that its denominator relies on the performance of existing MORL algorithms.... Moreover, determining accurate bounds requires running multiple MORL algorithms, increasing computational complexity.
>
> We would like to emphasise the importance and validity of the NHGR metric. First, the standard hypervolume is inherently scale-biased, as established in the paper. Although $HV_{\text{norm}}$ circumvents the scale bias issue in standard hypervolume measure, it is still biased in that it does not account for differences in maximally-achievable $HV_{\text{norm}}$ across environments/context. Intuitively, a context with more convex optimal pareto front would have higher maximally-achievable $HV_{\text{norm}}$ compared to one that is convex. If we were to aggregate (e.g. using IQM or optimality gap) the $HV_{\text{norm}}$ across contexts naively during evaluations, it would lead to bias towards contexts where there are more convex optimal Pareto fronts. As such, an agent which focusses on optimizing performances in environments with convex optimal Pareto fronts **only** would appear to generalize better than those who divide their learning to improve in every environment, even those with more concave optimal Pareto fronts. This goes against the **motivations of generalization in RL**. NHGR addresses this issue by estimating the maximal-achievable normalized hypervolume in each context using specialist agents, then evaluating the generalist agent’s performance as a ratio of this maximum. **This ensures that every evaluation context is fairly considered when assessing generalizability**. Estimation of the maximally-achievable hypervolume in each environment is unfortunately unavoidable when the reward ranges are unknown, which is typically the case. Such optimality threshold estimations is also common in SORL literature, e.g. SORL generalization benchmarks, multi-task RL, Atari 100k. Threshold estimations using specialist performances, as in NHGR, provides a more reliable way for estimating thresholds based on empirical evidence, compared to arbitrary estimates. As such, NHGR provides the **fairest (possible) pathway** for MORL generalization evaluation; it mitigates biases related to scale (of objectives) and environmental variations (in maximally-achievable hypervolume) to the best possible extent. Admittedly, determining accurate bounds requires running multiple MORL algorithms, but yet it is absolutely necessary for fair generalization evaluations. On the flip side, we argue that it **lends to the contributions of our work**. Specifically, the brunt of running these extensive specialist agent training is a computational cost borne by this work. This facilitates future research in MORL generalization as they can leverage the bounds and raw dataset established by us, thereby avoiding duplicated efforts.
>
> > 2-5 A potentially better approach would be to use the expected standard hypervolume across evaluation contexts, supplemented by the expected standard sparsity to address the limitations of relying solely on hypervolume.
>
> Thank you for the suggestion. However, the standard hypervolume is scale-biased and sparsity does not inform us about the maximum achievable hypervolume in each context, which are important considerations for evaluating MORL generalization emphasised above. On a side note, there have been recent discussions that the sparsity measure, initially introduced by PGMORL algorithm and became more prevalent in recent MORL literature, has some fundamental issues as an evaluation metric (see https://github.com/LucasAlegre/morl-baselines/pull/124). Ultimately, if future research wants to use other metrics, our software code already provides a few of them and offers the flexibility. However, the **scale-invariance** and **inter-environment fairness** properties of NHGR are strong reasons to retain it as a main measure of MORL generalization.

---

> ### Author Response · Authors · 2024-11-23
> **Response to Reviewer nVsL (Part 3/3)**
>
> ## Completeness and Contributions of our paper
>
> >  The overall contribution and novelty of this paper remain incomplete. While the paper highlights a promising research direction, it lacks a concrete solution for addressing the problem of MORL generalization. To enhance the paper’s completeness, the authors should propose and integrate their own method to tackle MORL generalization effectively.
>
> We respectfully disagree and appreciate the opportunity to clarify. Indeed, this paper highlights an important, yet overlooked, research direction for developing real-world agents. This significance has been acknowledged by other reviewers as well. We hope the reviewer empathize with the constraints of a conference-length publication. As the pioneering work in this promising area of research, there are so many potential areas that we can look into but ultimately, we decided it would be more meaningful to deliberately focus the paper’s scope in a way to **maximize its utility to future work investigating generalization + multi-objectivity**. To best advance this field of research, the paper provides (1) formalism for the generalization objectives of MORL agents, (2) a fair method for evaluating MORL generalizability, (3) a novel benchmark comprising 6 diverse domains with rich environment configurations, (4) extensive evaluations of current SOTA algorithms, (5) post-hoc discussions on the results and failure modes in current SOTA MORL methods as in Section 6.2 of the main body and Appendix B respectively. Most important of which, we provide software code for streamlining MORL generalization training and evaluation within our 6 domains, as well as contribute raw dataset from the extensive evaluation of 8 SOTA MORL algorithms on the 6 domains amassing >1000 GPU hours. These contributions sets the stage for future research seeking to improve MORL generalization by providing the necessary formalism, software tools, benchmarks, and insights to foster progress in this critical area. The baseline evaluations we provide in this paper significantly **reduces the computational and engineering-related friction for future research to begin investigating MORL generalization**. We must also point out that these contributions align significantly with the **Datasets & Benchmarks agendas of ICLR**, an overlooked (yet important for advancing research) subfield of conference publications in machine learning. The stance that extra algorithms to solve MORL generalization in order for this paper to be "complete" unfairly devalues the extensive contributions of this paper. Arguably, the theoretical contribution of a new algorithm and lengthy explanations accompanying it would form the scope of an entirely separate paper (succeeding this work) in itself.
>
> ---
>
> We sincerely hope the above responses have clarified your doubts regarding the paper and strengthened the case for our paper. If not, give us a holler and we are more than happy to further clarify.

---

> ### Comment · Reviewer_nVsL · 2024-11-26
>
> Thank you for the detailed responses to my questions!
>
> Could the authors discuss any potential strategy to mitigate computational challenges as the authors acknowledge that computational burden is inevitable when calculating NHGR (as mentioned in the response to 2-3 and 2-4)?

---

> ### Author Response · Authors · 2024-11-26
> **Response to Reviewer nVsL (Mitigating Computational Challenges for NHGR)**
>
> We sincerely thank you for your engagement in this discussion phase and the following should clarify your question:
>
> > Could the authors discuss any potential strategy to mitigate computational challenges as the authors acknowledge that computational burden is inevitable when calculating NHGR (as mentioned in the response to 2-3 and 2-4)?
>
> Here, we outline several strategies to address these challenges:
>
> * **early stopping**: the specialist performances usually take much shorter time steps than the generalist to converge in performance, so we often stopped training early. We mostly looked at the Weights & Biases plots for this (hypervolume, mean value for each objective, eum, etc.). For e.g., MO-Humanoid requires 10 millions steps for all generalist algorithms to converge but we were able to stop prematurely for most specialists at ~5 million steps.
> * **known lower/upper bounds**: when certain objectives have known lower/upper bounds (e.g., control cost with a maximum of 0 or forward velocity with a minimum of 0), we can use the known values and focus efforts on filling in the missing boundary values. Weaker or mid-range performance algorithms that are unlikely to contribute new Pareto points can be deprioritized, helping to avoid exhaustive search across all specialist algorithms.
> * **vectorized training**: it is challenging to implement vectorized environments for the multi-environment (generalist) training because of issues involving the underlying environment dynamics, such as the Mujoco engine. On the other hand, for specialist training to establish NHGR bounds, it is much easier to implement vectorized environments for specialist training because there is only a single fixed context.
> * **hyperparameter tuning**: there are already some established hyperparameters for existing MORL algorithms in the *single-environment* setting. We can use those hyperparameters as starting points where possible, and focus hyperparameter tuning efforts around those values for contexts that are untested/new.
> * **open-source effort**: we strongly advocate for a more open-source approach to advancing MORL generalization research. We hope the community will contribute pre-trained specialist performances via open-source, creating a shared repository of reusable results building on our own. This is particularly valuable because NHGR calculations can be performed post-hoc. Any updates to the hypervolume values of specialists automatically enable recalculation of NHGR scores for all generalist algorithms without requiring additional training; only the hypervolume values of the generalist agents need to be stored. This would could facilitate leaderboard-style comparisons of generalist MORL algorithms in the future, which we believe will be especially useful as the field increasingly acknowledges the inherently multi-objective nature of real-world generalization challenges.
>
> Ultimately, if future research wants to use other metrics, our software code already provides a few of them and offers the flexibility. However, the **scale-invariance** and **inter-environment fairness** properties of NHGR are strong reasons to employ it as a primary measure of MORL generalization, whenever possible (for e.g. domains that already have NHGR bounds established like ours). We would also like to share that we included a new section Appendix D.1 that discusses how to use an alternative MORL metric, EUM, in specific problems with clearly defined utility functions.
>
> ---
>
> We sincerely hope the above responses have improved your confidence in our paper. As always, we are happy to clarify any remaining questions you have! 😊

---

> > ### Comment · Reviewer_nVsL · 2024-11-27
> >
> > Thanks for the detailed explanations. I have adjusted my score accordingly.

---

> ### Author Response · Authors · 2024-11-27
> **Regarding current score**
>
> We appreciate your acknowledgment of the detailed explanations we’ve provided. However, we noticed that your revised score of 5 indicates that our work is still considered "marginally below the acceptance threshold". We would like to kindly seek clarification on whether the discussion has fully addressed your comments and resolved the concerns you raised earlier. You, and the other reviewers alike, have recognized that our pioneering work would be of valuable contribution to a critical field in RL research. We would like to understand if there are any aspects of our work that does not meet your standards of a publication-worthy paper.
>
> To further strengthen our case, we have posted an official comment summarizing the key contributions of our paper, along with links to our open-source code repository and a comprehensive response to all reviewers’ feedback. This comment highlights the significance and broader impact of our work and should address any remaining uncertainties. We respectfully invite you to review this summary at your convenience.
>
> Your feedback is invaluable, and we are committed to addressing any lingering concerns to ensure that our work serves as a solid foundation for future research into generalization in MORL.

---

> ### Author Response · Authors · 2024-12-01
> **Checking in**
>
> We hope this message finds you well. Perhaps you missed our previous message, but we would like to check in once more before the discussion period concludes, just to make sure we have addressed any of your lingering concerns. We kindly invite you to reconsider whether the current evaluation fairly captures the value of the contributions we’ve presented.
>
> As always, we remain committed to ensuring our work meets your standard of a publication-worthy paper. Thank you again for your engagement, thoughtful reviews, and voluntary contributions to advancing the research community. 😊

---

> ### Comment · Reviewer_nVsL · 2024-12-03
>
> I carefully reviewed the response and the revised PDF **again**. However, I still believe the work is not yet complete in its current form.
>
> A. Regarding the metric
>
> My concern from the comment 2-3 ("its denominator relies on the performance of existing MORL algorithms") pertains to cases where some algorithms perform poorly. If the performance of the combined specialists is low—whether due to the environment's complexity, insufficient fine-tuning, or inherent characteristics of a particular algorithm—the proposed metric might over-favor the evaluated algorithm because of a low denominator value.
>
> The authors might assume that in a benchmark setting, most MORL algorithms perform reasonably well. However, even in benchmark settings, this assumption is quite strong, especially considering future efforts to tackle more challenging benchmarks.
>
> As far as I understand, the authors have not explicitly addressed this issue in their response.
>
> Quick technical question: Can NHGR be defined using the ratio of unnormalized HV values? This might eliminate the need for bound calculations.
>
> B. Regarding the contribution
>
> First, I would like to acknowledge that I adjusted my original score after realizing I had underestimated the contribution. My comment was intended to highlight that even a simple method—even one that doesn't perform well—enhances the benchmark's completeness if it serves as the first algorithm developed for it. In similar cases, papers proposing new benchmarks often introduce their method (simple or otherwise) to demonstrate utility, as seen in works like Zhu et al. I do not deny the paper's whole contribution but suggest a straightforward way to improve its impact further. 😊
>
> Zhu, Baiting, Meihua Dang, Aditya Grover. Scaling Pareto-Efficient Decision Making via Offline Multi-Objective RL, ICLR 2023.

---

> > ### Author Response · Authors · 2024-12-03
> > **Regarding NHGR**
> >
> > We sincerely thank you for your comment and allowing us to better understand your uncertainty regarding the paper. We sincerely appreciate your time and we believe the following would address your raised concerns:
> >
> > > If the performance of the combined specialists is low—whether due to the environment's complexity, insufficient fine-tuning, or inherent characteristics of a particular algorithm—the proposed metric might over-favor the evaluated algorithm because of a low denominator value... considering future efforts to tackle more challenging benchmarks.
> >
> > We would like to first note that for the 6 domains in our benchmark, our specialists can perform reasonably well on them and using NHGR has significant benefits for fair evaluations compared to using standard hypervolume, as we thoroughly explained in our previous discussions. However, as you pointed out, it is important to consider the use of NHGR in more challenging benchmarks by future research where specialist approximations are inaccurate. Although this concern does not exactly pertain to the benchmark domains we have introduced, we are sincerely grateful to you for raising this aspect as it is important to address such considerations to ensure our work serves as a robust foundation for future research into generalization in MORL. Here, we seek to provide a reconcilation of the concerns you raised $$\rightarrow$$
> >
> > Indeed, in future benchmarks where even specialists can't perform well, it might lead to inaccurate NHGR approximations. Here we are faced with two possibilities: 1) use the average standard hypervolume (not NHGR) across all evaluation contexts as you suggested, or 2) stick with NHGR by relying on the current specialist performances as best approximation.
> >
> > Both approaches have trade-offs. (1) relies on arbitrary reference point assumptions and results in inter-objective scale biases and inter-environment bias. On the other hand, (2) would rely on suboptimal specialist approximations. Would you agree that providing results for both (1) and (2) would be a valuable approach in such cases? The standard hypervolume and NHGR are both metrics native to our software code and are tabulated during our training pipeline. Therefore, future research can definitely afford to plot both measures.
> >
> > Ultimately, as a pioneering work, our goal is not to impose rigid constraints on future research but to establish a flexible and well-justified starting point. In fact, we must point out that **the choice of performance metric has been an ongoing debate in multi-objective optimization (MOO) for decades**. This is because fundamentally, measuring the quality of Pareto fronts is more complex and multifaceted than unidimensional evaluations, and no single metric can suffice as the ultimate solution. NHGR, as discussed in our paper, draws inspiration from the hyperarea measure in MOO literature, which is well-regarded for its favorable properties. The goal of our pioneering work here is to lean on established studies in MOO and provide 2 well-motivated metrics that **seek to resolve many of the biases in current standard measures** (i.e. hypervolume and EUM). They are also the **most appropriate for our proposed benchmark domains**. Once again, we must emphasise that the software tool we contributed is flexible to accommodate the use of other metrics for future expansions into other benchmark domains.
> >
> > > Quick technical question: Can NHGR be defined using the ratio of unnormalized HV values? This might eliminate the need for bound calculations.
> >
> > Thank you for your question! We want to ensure we fully understand your suggestion—are you proposing using the ratio of unnormalized hypervolume values between the generalist and specialists, skipping the normalization step entirely? If so, that approach is indeed possible. However, one challenge we foresee is the introduction of inter-objective scale bias, which normalization is designed to mitigate. Additionally, the bound calculations in NHGR are derived directly from specialist performances, so collecting specialist performances naturally addresses this aspect simultaneously. Is there any motivation behind your suggestion that we failed to understand?  It’s possible we’ve missed an important perspective or context, and we’d love to explore it further. Our goal is to ensure that all suggestions for improving MORL generalization are thoughtfully considered in our work. 😊

---

> ### Author Response · Authors · 2024-12-03
> **Regarding the contribution**
>
> > First, I would like to acknowledge that I adjusted my original score after realizing I had underestimated the contribution. My comment was intended to highlight that even a simple method—even one that doesn't perform well—enhances the benchmark's completeness if it serves as the first algorithm developed for it.  In similar cases, papers proposing new benchmarks often introduce their method (simple or otherwise) to demonstrate utility, as seen in works like Zhu et al.
>
> Thank you for explaining your stance. We really appreciate that you were candid about your original underestimation of our contribution and for adjusting your score accordingly to account for it. We have sincere respect to you for that.
>
> Here, we seek to explain distinction between Zhu et al. (2023) and our work, and why it is much more difficult to introduce new methods in ours, if it does matter to your final evaluations of our work. Zhu et al. (2023) addresses **an already-established field of research**, *offline MORL*, where foundational tools and metrics are well-defined, making it more feasible to develop and include a new method. In contrast, our work aims to **establish a nascent area of research**, i.e. *MORL generalization*, which required us to take several foundational steps before even discussing algorithms. These include developing general formalisms for MORL generalization to standardize future discussions, proposing new metrics for quantifying generalization fairly, developing new software tools and pipelines from scratch to perform MORL generalization training, etc. All of these had to be established first before we could even discuss the proposed benchmark and the baseline evaluations. We also focussed on providing extensive post-hoc discussions, i.e. section 6 of the main body and Appendix A, B, C and D. These extensive discussions would really help lay a strong groundwork for future research into MORL generalization.
>
> We totally understand your perspective that proposing new methods would increase the impact of our work. Ultimately, more contributions the better. Beyond this work, we have already been looking into further research on algorithmic implementations to enhance MORL generalization. However, we still believe that the current work is complete on its own and given the constraints of a conference-length publications, we really don't think there is extra room to afford algorithmic improvements and their accompanying explanations within this work itself. We trust that you can empathise with that, given that you have already raised your score.
>
> ---
>
> Finally, we want to thank you again for being one of the 2 reviewers who actively engaged with us during the discussion period. Your constructive feedback has been instrumental in improving our manuscript, and we deeply value your contributions to the MORL research community. We hope you can look forward to future advancements in this field and perhaps even be interested in contributing. Once again, thank you.

---

> > ### Comment · Reviewer_nVsL · 2024-12-03
> >
> > Thank you for your detailed explanations.
> >
> > Regarding the metric, including standard metrics alongside NHGR, could mitigate the concern. However, since the issue I raised about NHGR still stems from its inherent definition, I am not entirely convinced of the metric's appropriateness.
> >
> > I may have missed some critical points, so please let me discuss this matter further with the other reviewers during the upcoming discussion period.
> >
> > Thank you for the kind responses!

---

> ### Author Response · Authors · 2024-12-04
> **Thank you Reviewer nVsL**
>
> We would like to acknowledge and sincerely thank you for the score raise. Upon discussions with you, it became clear that we as a pioneering work, we should have explicitly mentioned to readers that measuring the quality of Pareto fronts is more complex than unidimensional evaluations, and no single metric can suffice as the ultimate solution. Although NHGR and EUGR, have well-justified benefits over standard measures, they may be inaccurate in future more challenging domains where specialists do not perform well.
>
> As such, we are happy to share that we have included our discussions yesterday in our top-level comment, and we have made revisions to our manuscript to include evaluation results using hypervolume, NHGR, EUM, and EUGR. This should provide more informative and flexible options for future evaluations. We thank you for making our paper more comprehensive and balanced.

---

### Official Review · Reviewer_JmYT · 2024-10-23

**Soundness:** 2
**Presentation:** 3
**Contribution:** 2
**Rating:** 5
**Confidence:** 4

**Summary:**

This paper, based on the MuJoCo platform, presents a comprehensive set of parameter variations for different MORL (Multi-Objective Reinforcement Learning) testing scenarios. Additionally, it introduces a novel metric designed to standardize the evaluation of the Pareto front and hypervolume measurements. By establishing a wide range of parameterized benchmarks, this work evaluates recent state-of-the-art MORL algorithms across the proposed environment variations, highlighting the performance differences under diverse parameter settings. This serves as a valuable resource for researchers aiming to study the generalization capabilities of MORL algorithms, providing a flexible and well-defined platform for experimentation.

**Strengths:**

1. The paper introduces a substantial number of parameterized scenarios, significantly modifying standard MuJoCo test environments to induce pronounced changes in specific features. This extensive and detailed effort reflects a thorough approach in expanding the range of testing conditions.

2. It challenges the widely-used hypervolume metric by proposing a new metric based on hyperarea coverage, aimed at standardizing the quality assessment of the Pareto front.

3. Although there are some flaws in the writing, the overall readability is smooth, and the paper maintains a clear focus on its key contributions throughout the narrative.

**Weaknesses:**

1. **On the Use of Lebesgue Measure and Hypervolume:**
   The Lebesgue measure in two-dimensional space corresponds to calculating area. If we assume the solution set's Pareto front is a closed, well-defined curve, the area enclosed by the origin and the Pareto front can indeed be computed using the Lebesgue measure. In this specific two-dimensional case, where all solution sets are located in the first quadrant (i.e., all objective values are positive) and the origin is used as the reference point, the calculated hypervolume effectively becomes equivalent to the Lebesgue measure.

   It is important to note that the magnitude of the Lebesgue value is smaller and does not alter the comparative outcomes of optimization results. A smaller hypervolume (HV) will still lead to smaller two-point Lebesgue values. Although $HV_{norm} $ and $NHGR$ exhibit slightly higher penalty effects compared to the default in challenging scenarios, they do not provide additional benefits in terms of interpretability. Therefore, the proposed normalization does not seem to address the issue of result surges caused by outliers, as mentioned by the authors. Consequently, the introduction of $NHGR$ based on the hyper-area ratio appears to lack sufficient motivation.

2. **Regarding Metrics in Figure 3:**
   The two metrics presented in Figure 3 are actually different measurement dimensions of the same indicator. The mirrored images illustrate the overlap between these two metrics, which suggests they cannot be considered as introducing two new metrics.

3. **Multi-Objective Reward and Weight Vector Scalarization:**
   Multi-objective reward and weight vector scalarization is a well-established method. A substantial body of work has demonstrated the inefficiency of single scalar rewards in MORL optimization, such as in PGMORL and dynamic-weight MORL. Incorporating this concept as part of the experimental results does not significantly contribute to your experimental results, as it reflects existing knowledge in the field.

4. **Modifications to the Traditional Mujoco Environment:**
  In the original Mujoco environment, altering these values only requires adding instructions in the `step()` function, which is facilitated by the Mujoco simulation environment's API. The paper does not clearly indicate or demonstrate any additional code development or engineering improvements in this aspect.

5. **Efficiency of Benchmarking:**
   The benchmarking approach appears inefficient, as no new algorithm is proposed to compare against the state-of-the-art (SOTA).

**Questions:**

1. **Selection of Mujoco Environments:**
   Why were only certain environments from the Mujoco,i.e., only some of the two-ojective problems,  are selected as baselines for demonstration, while other important environments were omitted (problems of 3 or more objectives)? Including a broader range of environments would strengthen the validity and generalizability of the motivation.

2. **Modifications to the Traditional Mujoco Environment:**
   Is there any basis or reference for the numerical changes made to the standard Mujoco environment?

3.**Normalization range (NR):**
   Under what circumstances are NRs conducted, and why are they remain reliable despite changes in training or neural network parameters?

4.**Mujoco modifications:**
 Can you provide more details on any specific challenges encountered or innovations that has been made in implementing these modifications.

---

> ### Author Response · Authors · 2024-11-23
> **Response to Reviewer JmYT (Part 1/3)**
>
> We sincerely thank the reviewer for taking the time for writing such a detailed review amongst the other papers assigned for reviewing during this conference. The following clarifications will explain our stance and strengthen the case for our paper. We kindly request the reviewer's time to look through them. For all revisions to the manuscript, they are highlighted in $\color{teal}{\text{teal}}$.
>
> ---
>
> ## On the Use of Lebesgue Measure and Hypervolume
>
> > ...In this specific two-dimensional case, where all solution sets are located in the first quadrant (i.e., all objective values are positive) and the origin is used as the reference point, the calculated hypervolume effectively becomes equivalent to the Lebesgue measure.
>
> We believe there may have been some misunderstanding. The calculation of the hypervolume using Lebesque measure always corresponds to the volume of the Pareto front in the **positive direction** for all objectives, even those with negative scales, as per standard maximization goal of rewards in RL. **Prior to this work**, this is done by assuming an arbitrary reference point, a vector which each dimension corresponds to an **estimate** of the worst-possible return per objective because the true reward ranges are often unknown. For example, in MO-LunarLander, the reference point assumed in MO-Gymnasium [1] is [-101, -1001, -101, -101]. However, since the value for each dimension in the reference vector is arbitrary, and an exaggerated estimate of the worst-case reward in a specific dimension would lead to insignificant hypervolume differences with respect to that objective. For e.g., if we chose [-1001, -1001, -101, -101] as the reference point instead, then differences in reward obtained in the first objective dimension would become insignificant in hypervolume comparisons across algorithms. The arbitrary choice of reference point in hypervolume measures in current MORL literature results in unfair and problematic generalization evaluations if used naively. As such, hypervolume normalization in our paper, as in $HV_{\text{norm}}$, ensures that **each objective is given equal weight in the hypervolume calculation** by normalizing the scales of the rewards a priori with respect to min and max values calculated using specialist performances, before calculating hypervolume. This also **eliminates the need for reference point assumption**, we can use the origin vector since the range of objectives are now normalized to [0,1]
>
> [1] Florian Felten, Lucas N. Alegre, Ann Now´e, Ana L. C. Bazzan, El Ghazali Talbi, Gr´egoire Danoy, and Bruno C. da Silva. A toolkit for reliable benchmarking and research in multi-objective reinforcement learning. In Proceedings of the 37th Conference on Neural Information Processing Systems, 2023
>
> > A smaller hypervolume (HV) will still lead to smaller two-point Lebesgue values. Although $HV_{\text{norm}}$ and NHGR exhibit slightly higher penalty effects compared to the default in challenging scenarios, they do not provide additional benefits in terms of interpretability. Therefore, the proposed normalization does not seem to address the issue of result surges caused by outliers, as mentioned by the authors.
>
> We respectfully disagree and we appreciate the opportunity to clarify. First, the use of $HV_{\text{norm}}$ to circumvent the scale bias issue and eliminate the need for reference point assumption is established above. However, $HV_{\text{norm}}$ is still biased in that it **does not account for differences in maximally-achievable $HV_{\text{norm}}$ across environments/context**. Intuitively, an environment/context with more convex optimal pareto front would have higher maximally-achievable $HV_{\text{norm}}$ compared to one that is convex. If we were to aggregate (e.g. using IQM or optimality gap) the $HV_{\text{norm}}$ across contexts naively during evaluations, it would lead to bias towards contexts where there are more convex optimal Pareto fronts. As such, an agent which focusses on optimizing performances in environments with convex optimal Pareto fronts **only** would appear to generalize better than those who divide their learning to improve in every environment, even those with more concave optimal Pareto fronts. This goes against the **motivations of generalization in RL**. The NHGR metric accounts for this factor by first measuring the maximal achievable normalized hypervolume in each context via the specialists, then measure the agents performance in each context using the hypervolume ratio. This allows environments with concave Pareto fronts and lower achievable hypervolumes to be **equally weighted** against those with convex fronts during aggregations of performances across contexts. We hope the above explanation makes the motivation behind $HV_{\text{norm}}$ and NHGR clearer, and the importance of using NHGR in evaluating generalization performances.

---

> ### Author Response · Authors · 2024-11-23
> **Response to Reviewer JmYT (Part 2/3)**
>
> ## Miscellaneous, mainly regarding Mujoco Environments
>
> > The two metrics presented in Figure 3 are actually different measurement dimensions of the same indicator. The mirrored images illustrate the overlap between these two metrics, which suggests they cannot be considered as introducing two new metrics.
>
> This is likely because of the fact that none of the algorithms achieved optimality (as in NHGR=1) in any of the contexts, which leads to IQM and optimality gap to become almost symmetric counterparts. But ultimately, they are vastly different measures meant for different use cases which is why we included both of them. [2] makes clear the distinction between both measures. For posterity, we would like to clarify that in this paper, we only introduce $HV_{\text{norm}}$ and NHGR. IQM and optimality gap are two standardized metrics introduced by [2] for aggregating performances across different testing environments to circumvent statistical uncertainty inherent in RL, and are not introduced as new metrics in our paper.
>
> [2] Rishabh Agarwal, Max Schwarzer, Pablo Samuel Castro, Aaron C Courville, and Marc Bellemare. Deep reinforcement learning at the edge of the statistical precipice. Advances in neural information processing systems, 2021.
>
> > Under what circumstances are NRs conducted, and why are they remain reliable despite changes in training or neural network parameters?
>
> We believe there may have been some misunderstanding. The normalization ranges are determined from the specialist performances and it is only used for post-hoc evaluations, not during training. The estimated maximally-achievable performance in each objective dimension for each context serve as optimal thresholds for evaluation only, the generalist algorithms can choose any neural network parameters, or generalization approach to achieve these thresholds.
>
> > Why were only certain environments from the Mujoco,i.e., only some of the two-ojective problems, are selected as baselines for demonstration, while other important environments were omitted (problems of 3 or more objectives)?
>
> MO-Hopper has 3 objectives. Among the other standard Mujoco environments not introduced in this paper, there are the Reacher, Walker2D, Ant and Swimmer environment. However, all of them only have 2 components, which are the forward speed and control cost. within its hidden scalar reward function. We focussed on three primary environments MO-Hopper, MO-HalfCheetah, MO-Humanoid. Hopper has one of the smallest state-action dimensions amongst the standard Mujoco environments, and Humanoid has the most. We believe that 3 Mujoco domains, on top of the other 3 benchmark domains introduced in the paper, provides extensive enough coverage of generalization tasks, within our computational budget.
>
> > In the original Mujoco environment, altering these values only requires adding instructions in the step() function,... The paper does not clearly indicate or demonstrate any additional code development or engineering improvements in this aspect... Is there any basis or reference for the numerical changes made to the standard Mujoco environment? ... Can you provide more details on any specific challenges encountered or innovations that has been made in implementing these modifications.
>
> We respectfully disagree and appreciate the opportunity to clarify. Changing the underlying physics of the Mujoco engine such as agent weight and floor friction **do not** occur in the `step()` function. Rather, explicit changes to the Mujoco XML configurations need to be made in order to register additional elements such as floor friction and agent physics beyond the default values, which is an unintuitive process. The changes would then initialize in the Mujoco model. To alter the values during training, we have to find out which dimensions of the model are of interest, e.g. `friction = self.model.pair_friction[0,0]` and `masses = self.model.body_mass[1:]`. We must emphasise that these require **significant** engineering contributions and understanding of the Mujoco engine. The numerical changes used in our paper is mainly within reasonable neighbourhoods of the default values to maintain "learnability", and that also requires several trial-and-errors. For example, in MO-Hopper and MO-Cheetah, too low of a minimum friction value results in agent totally unable to move without slipping. We hope the above responses clarifies our contributions with respect to the Mujoco domains. We must remind the reviewer that our benchmark also includes 3 other domains (MO-LunarLander, MO-SuperMarioBros, and MO-LavaGrid), each requiring **significant** engineering efforts to establish. On top of that, we have developed a pipeline to train the MORL agents for generalization and post-hoc evaluations using $HV_{\text{norm}}$ and NHGR.

---

> ### Author Response · Authors · 2024-11-23
> **Response to Reviewer JmYT (Part 3/3)**
>
> ## Contributions of our paper
>
> > Multi-objective reward and weight vector scalarization is a well-established method. A substantial body of work has demonstrated the inefficiency of single scalar rewards in MORL optimization, such as in PGMORL and dynamic-weight MORL. Incorporating this concept as part of the experimental results does not significantly contribute to your experimental results, as it reflects existing knowledge in the field.
>
> We respectfully disagree and we appreciate the opportunity to make clear the contributions of our findings. Indeed, the limitations of scalar rewards are established in MORL, but **only** with respect to the single-environment setting. Specifically, the limitation expressed in current MORL literature prior to this work is that a SORL agent lacks the ability to satisfy different preferences over multiple objectives. The **consequences of scalar rewards in MORL generalization** (across multiple environments) has **never been explored**, especially given the absence of literature in MORL beyond static environments, and existing literature is not directly applicable. Our work is the first to train MORL agents in non-static/multiple environments for generalization. Our findings as in Section 6.2 provide counterintuitive motivations as to why the absence of multi-objectivity in current RL generalization literature is worrisome. As provided by the example in our paper, a SORL researcher may argue that if a stakeholder’s sole goal is for the agent to learn a **general** behavior for maximizing agent's forward movement (which seems like a single-objective problem) across different environments, there is no need to consider maximizing performances across different superfluous dimensions such as upward movement and control cost. However, our evaluations provide an counterintuitive finding that learning to maximize different dimensions, even those not related to the forward movement of the agent, can lead to discovery of behaviors that allow the agent to better generalize across different contexts (as shown in Figures 5 and 6). How multi-objectivity leads to diverse behaviors for generalization, even those that are seemingly single-objective problems, is a counterintuitive but novel empirical finding. Such finding would be of interest to the SORL generalization research community. And we can be certain of the novelty of this contribution because no other work in RL has studied generalization in multi-objective sequential decision making domains.
>
> > Efficiency of Benchmarking: The benchmarking approach appears inefficient, as no new algorithm is proposed to compare against the state-of-the-art (SOTA)
>
> We respectfully disagree and appreciate the opportunity to clarify. This paper highlights a promising research direction. On top of it, the paper provides (1) formalism for the generalization objective of MORL agents, (2) a formal method of evaluating it, (3) a novel benchmark comprising 6 diverse domains with rich environment configurations, (4) extensive evaluation of current SOTA algorithms, (5) post-hoc discussions of the results as in Section 6.2 of the main body and Appendix B. Most important of which, we provide software code for streamlining MORL generalization training and evaluation within our 6 domains, as well as contribute raw dataset from the extensive evaluation of 8 SOTA MORL algorithms on the 6 domains amassing >1000 GPU hours. These contributions align significantly with the **D&B agendas of the ICLR conference**. The stance that extra algorithms to solve MORL generalization in order for this paper to be "efficient" unfairly devalues the extensive and novel contributions of our paper. Arguably, the theoretical contribution of a new algorithm and lengthy explanations accompanying it would form the scope of an entirely separate paper in itself.
>
> ---
>
> We sincerely hope the above responses have clarified your doubts regarding the paper and strengthened the case for our paper. If not, give us a holler and we are more than happy to further clarify.

---

> > ### Comment · Reviewer_JmYT · 2024-11-27
> >
> > Thanks for the detailed response. I have adjusted the score accordingly. Good luck

---

> ### Author Response · Authors · 2024-11-27
> **Regarding current score**
>
> We appreciate your acknowledgment of the detailed explanations we’ve provided. However, we noticed that your revised score of 5 indicates that our work is still considered "marginally below the acceptance threshold". We would like to kindly seek clarification on whether the discussion has fully addressed your comments and resolved the concerns you raised earlier. You, and the other reviewers alike, have recognized that our pioneering work would be of valuable contribution to a critical field in RL research. We would like to understand if there are any aspects of our work that does not meet your standards of a publication-worthy paper.
>
> To further strengthen our case, we have posted an official comment summarizing the key contributions of our paper, along with links to our open-source code repository and a comprehensive response to all reviewers’ feedback. This comment highlights the significance and broader impact of our work and should address any remaining uncertainties. We respectfully invite you to review this summary at your convenience.
>
> Your feedback is invaluable, and we are committed to addressing any lingering concerns to ensure that our work serves as a solid foundation for future research into generalization in MORL.

---

> ### Author Response · Authors · 2024-12-01
> **Checking in**
>
> We hope this message finds you well. Perhaps you missed our previous message, but we would like to check in once more before the discussion period concludes, just to make sure we have addressed any of your lingering concerns. We kindly invite you to reconsider whether the current evaluation fairly captures the value of the contributions we’ve presented.
>
> As always, we remain committed to ensuring our work meets your standard of a publication-worthy paper. Thank you again for your engagement, thoughtful reviews, and voluntary contributions to advancing the research community. 😊

---

### Official Review · Reviewer_yE1f · 2024-10-31

**Soundness:** 2
**Presentation:** 3
**Contribution:** 2
**Rating:** 6
**Confidence:** 4

**Summary:**

This paper investigates a very important problem of generalization in multi-objective reinforcement learning (MORL), an area that has received limited attention compared to single-objective RL. The main contribution of the paper includes a novel testbed featuring diverse MORL domains with different contexts, which provides a systematic evaluation framework for generalization in MORL. The paper also presents a comprehensive analysis of state-of-the-art MORL methods (Envelope, GPI, CAPQL, PGMORL, MORL/D) using normalized hypervolume generalization ratio. Overall, the paper provides a foundation for understanding how MORL algorithms generalize across different environmental contexts and parameters, addressing a crucial gap in the current literature.

**Strengths:**

* The paper contributes a novel testbed featuring various multi-objective RL environments with parameterized environment configurations, facilitating valuable research in MORL generalization.
* The proposed framework for evaluating generalization in MORL using normalized hypervolume generalization ratio is sound and promising.
* The paper provides reasonable coverage of relevant literature and effectively justifies its claims with thorough experimental evaluations.
* The paper is generally well-written and easy to follow.

**Weaknesses:**

* The proposed testbed appears to be a straightforward extension of MORL environments to generalization with limited variations, lacking clear evidence of full interpolation/extrapolation properties across context changes.
* The evaluation lacks analysis of important MORL metrics such as sparsity, expected utility, and cardinality, which limits the comprehensive understanding of algorithm performance. For instance, while hypervolume measures Pareto coverage, it does not address how spread the solutions are within the Pareto front, which is an important concept in utility-based approaches. Similarly, the expected utility metric is essential as it directly reflects an agent’s ability to maximize total utility, which is the ultimate goal in utility-based approaches.
* The paper's focus on utility-based approaches with limited discussion of different utility or scalarization functions restricts the broader applicability. For instance, for linear scalarization functions, the resulting Pareto front is usually convex while for non-linear scalarization functions, the resulting Pareto front may have concave regions. It is unclear which class of scalarization method is used and how different classes affect generalization in MORL.
* The empirical evaluation is limited to environments with a small number of objectives (maximum 4) which raise questions about scalability and generalizability. Testing in environments with higher objective counts, such as the Fruit Tree Navigation environment (5-7 objectives) by Yang et al. (2019) or the MO-highway environment from the mo-gymnasium API, which can be configured for more objectives, would strengthen the paper.

**Questions:**

1. How does the proposed framework handle cases where utility functions are unknown, non-linear, or non-monotonic?
2. What specific scalarization functions were used in the MORL algorithms, and how were weight vectors initialized in methods like the Envelope algorithm?
3. Why were other MORL metrics such as sparsity, cardinality, and expected utility not considered during evaluation?
4. How do MORL methods generalization affected by varying numbers of objectives in underlying environments?
5. Can the authors explain the discrepancy in CleanRL SAC performance in default hopper environment which achieved 3000 episodic return vs the one reported in Figure 5 (left most plot)? Could this because of the hyperparameter? If yes, then a well-tuned SAC may outperform the MORL methods.


Minor comments:

In Definition 1, should the reward be represented as a vector in bold notation for consistency with Section 2?

The related work should be placed in the main paper. This can be done by moving the detailed environment descriptions to the appendix for better space utilization.

---

> ### Author Response · Authors · 2024-11-27
> **Response to Reviewer yE1f (Part 1/3)**
>
> We sincerely thank the reviewer for their detailed and thoughtful feedback, especially considering the volume of papers assigned for this conference. We value the opportunity to address your concerns and clarify key points, and we have revised our manuscript to incorporate these clarifications, with changes highlighted in $\color{teal}{\text{teal}}$. Below, we address each question and comment in turn.
>
> ---
> ## Clarification
>
> First, we hope to get the clarification questions out of the way before responding to the other points regarding weaknesses of the paper.
>
> > What specific scalarization functions were used in the MORL algorithms, and how were weight vectors initialized in methods like the Envelope algorithm?
>
> The MORL algorithms primarily employ linear scalarization with weights summing to unity. This choice is not specific to our implementation but is instead a limitation of these algorithms, which rely on linear utilities to simplify learning. For instance, both CAPQL and GPI-LS depend on the assumption of linear utilities for their operation. We have clarified this in Appendix Section A. Weights are uniformly sampled across the weight space at the beginning of every new episode or replay.
>
> > How do MORL methods generalization affected by varying numbers of objectives in underlying environments?
>
> Intuitively, as the number of objectives increases in the underlying problem, the learning of the agent becomes more complex and it would be harder to generalize and perform optimally across the increased number of objectives over all environments. However, the degree of difficulty also depends on the level of conflict between objectives and the problem's inherent complexity. For example, while MO-LunarLander has four objectives, algorithms perform better on it than MO-LavaGrid, which has only two objectives. Generalization difficulty is not solely determined by the number of objectives but also by how these objectives interact and the structure of the environment. Therefore, no definitive conclusion can be made.
>
> > Testing in environments with higher objective counts, such as the Fruit Tree Navigation environment (5-7 objectives) by Yang et al. (2019) or the MO-highway environment from the mo-gymnasium API, which can be configured for more objectives, would strengthen the paper.
>
> Thank you so so much for the suggestion, we have been looking for environments with many objectives, can't believe we missed out Fruit Tree Navigation! It's so easy to add variations too, we just needed to set the values at the leaf nodes. We have implemented Fruit Tree Navigation within our open-sourced codebase (link above) and are running the baselines. However, given computational limitations within our lab during this ICLR rebuttal period, we sincerely apologise if we can't provide the results before the discussion period ends. 😔
>
> > Can the authors explain the discrepancy in CleanRL SAC performance in default hopper environment which achieved 3000 episodic return vs the one reported in Figure 5 (left most plot)? Could this because of the hyperparameter? If yes, then a well-tuned SAC may outperform the MORL methods.
>
> The discrepancy arises because, in Figure 5, CleanRL SAC is trained using domain randomization to generalize across **all** contexts, whereas the CleanRL open-sourced results refer to SAC trained to specialize on a **single** context (i.e., the default Hopper environment). Figure 5 compares the $f_{\text{SORL}}$ scalarized performance of MORL algorithms and SAC, both trained via domain randomization. While SAC explicitly trains to optimize $f_{\text{SORL}}$, the MORL methods optimize individual objectives independently. Interestingly, our results demonstrate that MORL approaches often achieve superior $f_{\text{SORL}}$ performance by learning the objectives separately rather than directly optimizing the scalarized objective, as SAC does.
>
> The training budget and hyperparameters for all the domains have now been included in Appendix E and F. Specifically, for MO-LavaGrid, MO-LunarLander, MO-Hopper, MO-HalfCheetah, and MO-Humanoid, all our policy and value functions are multi-layer perceptrons (MLPs) with four hidden layers of 256 units each. This is arguably on the larger end of most RL algorithms, at least within these Gymnasium-based environments (for reference, CleanRL only uses 2 hidden layers for Mujoco environments). For MO-SuperMarioBros, we use a NatureCNN network + two hidden layers of 512 units each. All our replay buffer sizes are 1 million transitions, except in MO-SuperMarioBros where it is 100,000 because storing pixel images is rather memory intensive. As such, we can assure that the hyperparameters are non-issue.
>
> > In Definition 1, should the reward be represented as a vector in bold notation for consistency with Section 2?
>
> Appreciate the sharp eye, fixed!
>
> > The related work should be placed in the main paper.
>
> Thanks for the suggestion, we have revised as such!

---

> ### Author Response · Authors · 2024-11-27
> **Response to Reviewer yE1f (Part 2/3)**
>
> ## Discussion on other metrics and Generality of the Hypervolume metric
>
> > The paper's focus on utility-based approaches with limited discussion of different utility or scalarization functions restricts the broader applicability. For instance, for linear scalarization functions, the resulting Pareto front is usually convex while for non-linear scalarization functions, the resulting Pareto front may have concave regions. It is unclear which class of scalarization method is used and how different classes affect generalization in MORL.
>
> We believe that there has been some misunderstanding and we appreciate the opportunity to clarify. Our evaluations is **not restricted to and does not assume** any utility function. Rather, it is the algorithms themselves that often assume certain utility types to simplify the problem and facilitate learning (e.g., via Q-learning). But this does not imply that benchmark environments or human preferences are inherently represented by known utility functions
>
> In this paper we wanted to provide a **general** metric that measures the generalizability of any MORL approach. We make no assumptions of the underlying MORL approach, i.e. whether it is axiomatic or whether it depends on linear/non-linear scalarization. Therefore, using hypervolume is justified—a Pareto front that maximizes hypervolume will also maximize expected utility of any monotonic utility function, but not the other way around.
>
> However, in specific case where there is prior knowledge of the true utility function, EUM is indeed a valuable tool for evaluation. Appendix D.1 provides a detailed discussion of how EUM can be applied for MORL generalization evaluations in these cases. Additionally, we introduce the Expected Utility Generalization Ratio (EUGR), a measure motivated by NHGR, to enable meaningful comparisons when utility priors are available. This section also includes plots illustrating results obtained using EUM and EUGR. Extended metric discussions and results can be find throughout Appendix D.
>
> > How does the proposed framework handle cases where utility functions are unknown, non-linear, or non-monotonic?
>
> With the previous point, these scenarios can be be summarised as:
> 1. **Unknown utility function**: Hypervolume (NHGR) is a general metric that accounts for any monotonically increasing utility function
> 2. **Known linear/non-linear utility function**: Use expected utility (EUGR) to perform generalization evaluations
>
> Non-monotonic utility function contradicts the notions of reward -- getting more reward for an objective should not decrease a user's utility as long as it does not result in a decrease in reward for another. To our best knowledge, MORL literature do not consider such cases.
>
> > The evaluation lacks analysis of important MORL metrics such as sparsity, expected utility, and cardinality, which limits the comprehensive understanding of algorithm performance...Similarly, the expected utility metric is essential as it directly reflects an agent’s ability to maximize total utility, which is the ultimate goal in utility-based approaches.
>
> We recently realized that there have been discussions that the sparsity measure, initially introduced by PGMORL algorithm and became more prevalent in recent MORL literature, has some fundamental issues as an evaluation metric (see https://github.com/LucasAlegre/morl-baselines/pull/124). While we (the MORL research field) look into rectifying, we think it's better not to include it in the paper to prevent further propagation.
>
> As mentioned earlier, Appendix D provides discussion on EUM and also introduces new EUM-based measures of generalization (i.e. EUGR) and results. Cardinality is also provided as a measure in our software code, with results available on Weights and Biases and also as CSVs in our code repository.

---

> ### Author Response · Authors · 2024-11-27
> **Response to Reviewer yE1f (Part 3/3)**
>
> ## Contributions of our paper
>
> > The proposed testbed appears to be a straightforward extension of MORL environments to generalization with limited variations, lacking clear evidence of full interpolation/extrapolation properties across context changes.
>
> We believe that this statement undervalues the contributions of our work, and we appreciate the opportunity to clarify. While our testbed may appear to be a simple extension of existing MORL environments, it involves substantial engineering efforts and careful design choices to support meaningful generalization studies. For instance, working with the Mujoco engine requires understanding of its XML configurations to implement context variations. Designing these benchmarks also require many trial-and-error to ensure the environments remain "learnable." For example, in MO-Hopper and MO-LunarLander, setting the friction/gravity values inappropriately would render the task unlearnable. MO-SuperMarioBros, previously limited to stage 1-1, was expanded to encompass 32 distinct stages. Additionally, MO-LavaGrid is an entirely new environment we designed to test MORL generalization across reward functions.
>
> A lot of thought also went into the design of the benchmark to ensure it captures the complexity of real-world generalization challenges. We deliberately ensured coverage of **state-space, dynamics, and reward-function variations** in our domains, each targeting a distinct dimension of generalization, as outlined in [1]. Also, [1] notes that *"...dynamics variation is only tackled in limited settings and reward-function variation is very under-studied. These stronger forms of variation are still likely to appear in real-world scenarios, and hence should be the focus of future research"*. By introducing dynamics variations, our environments directly address this gap while also contributing to the challenge of sim-to-real transfer, an important area of ongoing research. Additionally, by adding a multi-objective dimension into the generalization problem, our benchmark aligns more closely with the complexities of real-world, multi-objective scenarios, further increasing its relevance.
>
> Extrapolation studies form a separate line of inquiry and are often addressed in distinct research (e.g., SORL generalization versus zero-shot generalization). Still, our software code and testbed lays the groundwork for such investigations and we show some preliminary study on zero-shot generalization in Fig. 4a. Finally, Section 6.2 and Appendix D.3 include a counterintuitive yet significant finding on the **importance of interpolation** in our proposed domains. Specifically, we show that under all our domains, MORL agents demonstrate better interpolation capabilities when objectives are learned independently rather than relying on scalarized learning, underlying the importance of incorporating multi-objectivity when training generally-capable agents.
>
> [1] Robert Kirk, Amy Zhang, Edward Grefenstette, and Tim Rockt¨aschel. A survey of zero-shot generalisation in deep reinforcement learning. J. Artif. Int. Res., 76, May 2023
>
> ## Reiterating Contributions
>
> As the pioneering work in this promising area of research, there are so many potential areas that we can look into but ultimately, we decided it would be more meaningful to deliberately focus the paper’s scope in a way to **maximize its utility to future work investigating generalization + multi-objectivity**. To best advance this field of research, the paper provides (1) formalism for the generalization objectives of MORL agents, (2) fair methods for evaluating MORL generalizability, (3) a novel benchmark comprising 6 diverse domains with rich environment configurations, (4) extensive evaluations of current SOTA algorithms, (5) post-hoc discussions on the results and failure modes in current SOTA MORL methods, as well as extensive discussions on future areas of research for improving MORL generalization.
>
> Most importantly, we provide software code for streamlining MORL generalization training and evaluation within our 6 domains, as well as contribute raw dataset from the extensive evaluation of 8 SOTA MORL algorithms on the 6 domains amassing >1000 GPU hours. These contributions sets the stage for future research seeking to improve MORL generalization by providing the necessary software tools, benchmarks, and insights to foster progress in this critical area. The baseline evaluations we provide in this paper significantly reduces the computational and engineering-related friction for future research to begin investigating MORL generalization. We must also point out that these contributions align significantly with the **Datasets & Benchmarks agendas of ICLR**, an overlooked yet important pillar of machine learning.
>
> ---
> We sincerely hope the above responses have clarified your doubts regarding the paper and strengthened the case for our paper. If not, give us a holler and we are more than happy to further clarify! 😊

---

> ### Author Response · Authors · 2024-12-01
> **Checking in**
>
> We hope this message finds you well. We noticed we have yet to hear from you during this discussion period. As the discussion period nears it end, we just want to make sure that we have addressed any remaining concerns you may have. Our goal is to ensure that our work serves as a strong foundation for future research into generalization in MORL. Your thoughtful insights have been crucial in shaping the trajectory of our work, and we would like to sincerely thank you for your valuable contributions.

---

### Official Review · Reviewer_8Wih · 2024-11-01

**Soundness:** 2
**Presentation:** 3
**Contribution:** 2
**Rating:** 6
**Confidence:** 4

**Summary:**

This paper studies generalization in the context of multi-objective reinforcement learning (MORL), which extends standard RL to the case in which the reward function is a vector containing multiple conflicting objectives. In particular, the paper defines the problem as a contextual multi-objective MDP, in which a context variable defines changes in the MDP’s state-transition and reward functions. The goal of an agent is to learn a set of Pareto optimal policies that generalize over different contexts. Next, the paper introduces a set of environments that extends existing benchmark environments in MO-Gymnasium to the case in which the dynamics and the state space can be modified. The paper evaluates several state-of-the-art MORL algorithms in the introduced benchmark and shows that existing algorithms often do not generalize well to different contexts.

**Strengths:**

- This is the first work, to the best of my knowledge, that studies MORL under environments with changes in its dynamics/state-transition function.
- The introduced benchmarks are potentially useful to the community for designing novel MORL algorithms that can tackle problems in which the MOMDP's dynamics can change.

**Weaknesses:**

- Since the main contribution of the paper is the testbed containing new extensions of MORL benchmarks and an extensive evaluation of existing algorithms, it is possible that this paper would be more suitable to the Dataset & Benchmarks track than the main track of the conference.
- The theoretical contributions are limited to the definition of a contextual MOMDP, which follows directly from previously studied formalisms of MOMDP and contextual MDPs, and the definition of a normalized evaluation metric for evaluating agents in this formalism.
- The paper requires clarifications regarding the mathematical formulation of the problem, and its implications on the evaluation metrics used and the experimental results (see below).
- In particular, it is not clear if the existing MORL algorithms are performing poorly due to their limitations or if the proposed benchmark is actually impossible to solve with optimality (i.e., obtaining NHGR=1). For instance, in the Mujoco environments, it is impossible for an agent to act optimally w.r.t. all possible friction coefficients without having any form of access to the current coefficients.

**Questions:**

Below, I have questions and constructive feedback to the authors:

1) Regarding Definition 3, what does it mean for a policy to be “generalized across contexts”? There are two options here: (i) learn a single policy that maximizes its expected value over contexts, possibly being suboptimal because a policy that is optimal to every context simultaneously likely does not exist; or (ii) learn a policy that is conditioned on the context, and then can adapt its behavior depending on the context. I suggest discussing and clarifying this aspect of the problem defined in Section 3. In case (ii), the environments introduced should contain the context as part of the state space.

2) In Definition 4, I suggest defining how the Pareto front is normalized and mentioning the range of the normalization, e.g., [0,1].

3) Given that, in Definition 3, the authors are restricting to linear utility problems (which induce convex Pareto Fronts), it is worth noting that hypervolume is a metric that considers Pareto-optimal points in concave regions of the Pareto front, which are not useful for increasing the expected utility. For example, if we have an optimal convex Pareto front and we add a point in a concave region, this point would increase the hypervolume but would not increase the expected utility. For this reason, I suggest focusing on expected utility metrics (following the utility-based approach) instead of using hypervolume, which is an axiomatic metric. Currently, it is contradictory that the paper advocates for utility-based approaches, but mainly uses an axiomatic metric for evaluation. I suggest the authors either justify their use of hypervolume in this context or adopt metrics more consistent with their utility-based approach (e.g., normalized expected utility).

4) It is unclear whether (in both the problem formulation and in the proposed benchmark) the context is part of the observation, i.e., whether agents know which context they are in. If the agents do not have access to the context, it is expected that generalist agents do not perform well. In general, a different policy is required to act optimally with respect to each context, and learning a policy that optimizes for the average context will be probably suboptimal to every context (unless contexts are too similar). Hence, the NHGR metric in Definition 5 would never be equal to 1, even assuming access to a perfect algorithm that learns the optimal policy in Definition 3. I suggest the authors clarify whether perfect generalization (NHGR=1) is theoretically possible in these environments.

5) Regarding the MO-SuperMarioBros, the paper mentions that “There are a total of 32 possible stages”. Is that correct? Based on the Appendix, there are 8 stages.

6) In Section 6.1, what was the training budget in terms of environment interactions for the specialist and generalist agents? Since generalist agents have to learn to solve multiple contexts instead of only one, it would be fair that they are given sufficient time to learn them. Otherwise, would it be possible to explain the results solely on the fact that the generalist agents did not have enough training time?

7) I also suggest discussing the hyperparameters used for the algorithms in the experiments. For instance, would it be necessary to use larger neural networks or larger replay buffers when we train an agent to optimize for many contexts simultaneously? It is not clear whether the algorithms do not perform well due to a lack of hyperparameter turning.

Minor: “long-term discounted reward” - > long-term discounted sum of rewards.

**Details Of Ethics Concerns:**

I identified no ethical concerns that need to be addressed in this paper.

---

> ### Author Response · Authors · 2024-11-26
> **Response to Reviewer 8Wih (Part 1/3)**
>
> We sincerely thank the reviewer for taking the time to write such a detailed review amongst the many other papers assigned for reviewing during this conference. The following clarifications will explain our stance and strengthen the case for our paper. We kindly request the reviewer's time to look through them. For all revisions to the manuscript, they are highlighted in $\color{teal}{\text{green}}$.
>
> ---
>
> ## Clarifications
>
> First, we hope to get the minor clarification questions out of the way before addressing the main "weaknesses" in subsequent parts.
>
> > 2. In Definition 4, I suggest defining how the Pareto front is normalized and mentioning the range of the normalization, e.g., [0,1].
>
> Thank you for the suggestion, we have addressed this in the revised manuscript.
>
> >  5. Regarding the MO-SuperMarioBros, the paper mentions that “There are a total of 32 possible stages”. Is that correct? Based on the Appendix, there are 8 stages.
>
> The agents are trained on 31/32 stages (Stage 3-3 is excluded to test for zero-shot generalization). The 8 stages are the evaluation environments/contexts. We evaluate on only on 8/32 of the stages to keep the runtime within reasonable limits. Regardless, thanks for highlighting this confusion, we have made this clearer in the revised manuscript (Appendix E and F).
>
> > 6. In Section 6.1, what was the training budget in terms of environment interactions for the specialist and generalist agents? ... would it be possible to explain the results solely on the fact that the generalist agents did not have enough training time?
>
> The training budget and hyperparameters for all the domains have now been included in Appendix E and F. Compared to [1] which only trains on static environments, our training times for all MORL algorithms are **much longer** in all our generalization domains. We select the training budget based on however long it takes for **all MORL algorithms to converge in performance**. We mainly look at the Weights & Biases plots for this. Under Table 2 of Appendix E, you can see that the training duration for each environment differs; the harder ones with bigger state-action spaces are typically longer. As such, we can guarantee you that the training budget is not a factor for the poor performances.
>
> [1] Florian Felten, Lucas N. Alegre, Ann Now´e, Ana L. C. Bazzan, El Ghazali Talbi, Gr´egoire Danoy, and Bruno C. da Silva. A toolkit for reliable benchmarking and research in multi-objective reinforcement learning. In Proceedings of the 37th Conference on Neural Information Processing Systems, 2023
>
> > 7. I also suggest discussing the hyperparameters used for the algorithms in the experiments. For instance, would it be necessary to use larger neural networks or larger replay buffers when we train an agent to optimize for many contexts simultaneously?
>
> The training budget and hyperparameters for all the domains have now been included in Appendix E and F. Specifically, for MO-LavaGrid, MO-LunarLander, MO-Hopper, MO-HalfCheetah, and MO-Humanoid, all our policy and value functions are multi-layer perceptrons (MLPs) with **four hidden layers of 256 units each**. This is arguably on the larger end of most RL algorithms, at least within these Gymnasium-based environments (for reference, CleanRL only uses 2 hidden layers for Mujoco environments). For MO-SuperMarioBros, we use a NatureCNN network + two hidden layers of 512 units each. All our **replay buffer sizes are 1 million transitions**, except in MO-SuperMarioBros where it is 100,000 because storing pixel images is rather memory intensive. As such, we can assure that the network and replay buffer sizes are non-issue.
>
> > Minor: “long-term discounted reward” - > long-term discounted sum of rewards.
>
> Appreciate the sharp eye, fixed!

---

> ### Author Response · Authors · 2024-11-26
> **Response to Reviewer 8Wih (Part 2/3)**
>
> ## Choice of metric + Reiterating contributions of paper
>
> > 3. Given that, in Definition 3, the authors are restricting to linear utility problems (which induce convex Pareto Fronts) ...it is contradictory that the paper advocates for utility-based approaches, but mainly uses an axiomatic metric for evaluation.
>
> We believe that there has been some misunderstanding, and we appreciate the opportunity to clarify. Definition 3 **does not restrict to linear utility problems**. In the equation $\pi_{\mathbf{w}} = arg \max\_{\pi \in \Pi} \mathbb{E}\_{c \sim p(c)} \left[u_{\mathbf{w}}(\mathbf{v}_c^\pi)\right]$, $u(\cdot)$ is an arbitrary utility function that can be linear or non-linear. While utility-based MORL algorithms often employ linear scalarization to simplify the problem and facilitate learning (e.g., via Q-learning), this does not imply that benchmark environments or human preferences are inherently represented by linear utility functions. In practice, the true utility function of stakeholders, if any, is often unknown.
>
> Therefore, using hypervolume instead of expected utility as the primary measure is crucial—a Pareto front that maximizes hypervolume will also maximize expected utility of **any** monotonic utility function, but not the other way around. Using the expected utility metric (EUM) requires assuming a specific utility function for calculations, which significantly limits the generality and applicability of our generalization evaluations, especially since the true utility function is often unknown. Our goal here is to have a **general metric** that measures the generalizability of any MORL approach, be it axiomatic, linear or non-linear scalarization, and hypervolume would be most appropriate. Hypervolume also possesses many good mathematical properties as a cherry on top, and it is rooted deeply in multi-objective optimization (MOO) literature.
>
> > Since the main contribution of the paper is the testbed containing new extensions of MORL benchmarks and an extensive evaluation of existing algorithms, it is possible that this paper would be more suitable to the Dataset & Benchmarks track than the main track of the conference. The theoretical contributions are limited to the definition of a contextual MOMDP, which follows directly from previously studied formalisms of MOMDP and contextual MDPs, and the definition of a normalized evaluation metric for evaluating agents in this formalism.
>
> This paper was not meant to be a theoretical paper, and we apologize if we gave a different impression. We sincerely appreciate that you recognized the dataset and benchmark contributions of the paper. Actually, that was the primary motivation behind this paper and we would like to ask for another opportunity to make a case for our paper:
>
> As the pioneering work in this promising area of research, there are so many potential areas that we can look into but ultimately, we decided it would be more meaningful to deliberately focus the paper’s scope in a way to **maximize its utility to future work investigating generalization + multi-objectivity**. To best advance this field of research, the paper provides (1) formalism for the generalization objectives of MORL agents, (2) a fair method for evaluating MORL generalizability, (3) a novel benchmark comprising 6 diverse domains with rich environment configurations, (4) extensive evaluations of current SOTA algorithms, (5) post-hoc discussions on the results and failure modes in current SOTA MORL methods as in Section 6.2 of the main body and Appendix B respectively. Most important of which, we provide software code for streamlining MORL generalization training and evaluation within our 6 domains, as well as contribute raw dataset from the extensive evaluation of 8 SOTA MORL algorithms on the 6 domains amassing >1000 GPU hours. These contributions sets the stage for future research seeking to improve MORL generalization by providing the necessary formalism, software tools, benchmarks, and insights to foster progress in this critical area. The baseline evaluations we provide in this paper would significantly **reduces the computational and engineering-related friction for future research to begin investigating MORL generalization**.
>
> There is no separate D&B track for ICLR, so we submitted it to the main conference. But it appears that D&B is one of the listed agendas of ICLR in its call for papers, so thank you for highlighting that! Datasets and benchmarks are crucial for advancing ML research. We believe our work catalyzes discussions in an extremely important direction of RL research that is necessary for developing agents capable of tackling the complexities of real-world, multi-objective scenarios. We hope our clarifications would allow you to view the paper in a more positive light.

---

> ### Author Response · Authors · 2024-11-26
> **Response to Reviewer 8Wih (Part 3/3)**
>
> Finally, it appears that your primary concern with our paper is regarding the benchmark and whether the proposed domains are actually "solvable" (NHGR = 1). We thank you for raising this concern, and we hope that addressing it would strengthen the case for our paper. Whatever we write below is also included **under Appendix C in more detailed form**.
>
>  ---
>
> ## Principle of Unchanged Optimality
>
> It appears that you are referring to *The Principle of Unchanged Optimality* [2]. This principle asserts that, for a domain to support generalization, it should provide **all necessary information such that a policy optimal in every context can exist**. This principle is often overlooked by generalization literature, but arguably extremely important, so we thank you for your astute insight. We agree that clarifying this principle within our paper will significantly strengthen our paper and set a strong foundation for MORL generalization research ahead.
>
> Indeed, *if* this principle was to be violated, generalist agents would be fundamentally unable to achieve an NHGR score of 1, as they could never match the performance of specialists across all contexts. We appreciate the opportunity to explain why *The Principle of Unchanged Pareto Optimality* is upheld in **all** the domains of our MORL generalization benchmark, thereby supporting its validity for measuring MORL generalization.
>
> For MO-SuperMarioBros, each observation provides sufficient information for determining the optimal action at every time step (e.g. immediate coins, enemies, bricks are visible). Similarly, in MO-LavaGrid, the placement of lava and goals is fully observable in each observation. Furthermore, as described in Section 5, we concatenate the reward weights of each goal with the agent’s observations, a deliberate choice to ensure that necessary information about the context, specifically the current reward function, is provided to the agent for optimal planning.
>
> For MO-Hopper, MO-HalfCheetah, MO-Humanoid and MO-LunarLander, each context varies in environment dynamics, such as gravity and friction. However, the agent’s observations only include the positions and velocities of the robot’s joints, making it impossible to infer the environment dynamics (context) or determine optimal actions from a single time step. We can understand why you are concerned about this. However, necessary information about the environment dynamics is **inferrable when the agent considers its state-action history**. Consequently, when the context is hidden, the optimal policies (as in optimal across all contexts) in these domains are inherently non-Markovian, requiring either memory-based/recurrent policies or regression over state-action history buffers.
>
> Therefore, we must stress that **The Principle of Unchanged Optimality can indeed be upheld in all our proposed domains**, depending on the provided algorithms. Specifically, perfect generalization (NHGR=1), or at least $\epsilon$-perfect generalization, would be theoretically possible in these environments. The problem, however, lies with the **immaturity of current MORL algorithms for generalization** which largely assume Markovian policies and are not adapted for generalization across multiple environments. But this is expected since we are the first work to consider MORL outside more than one environments.
>
> The primary objective of this paper is to provide future research with extensive baselines using current MORL methods, equipping the field with better insights into which approaches to build upon. We anticipate that future work on MORL generalization, building on our paper, will explore creative adaptations of existing SORL generalization methods to MORL algorithms. And that is exactly why we implemented our open-source software as a primary motivation–to facilitate algorithmic adaptations and simplify benchmarking. As a starting point, we recommend examining methods proposed in [3,4,5]. We would like to refer you to Section B and C of the appendix for more detailed discussions on limitations in current MORL algorithms, *The Principle of Unchanged Optimality*, and possible future work.
>
> [2] Alex Irpan and Xingyou Song. The principle of unchanged optimality in reinforcement learning generalization, 2019
>
> [3] Wenhao Yu, Jie Tan, C. Karen Liu, and Greg Turk. Preparing for the unknown: Learning a universal policy with online system identification, 2017.
>
> [4] Xue Bin Peng, Marcin Andrychowicz, Wojciech Zaremba, and Pieter Abbeel. Sim-to-real transfer of robotic control with dynamics randomization, 2018.
>
> [5] Charles Packer, Katelyn Gao, Jernej Kos, Philipp Kr¨ahenb¨uhl, Vladlen Koltun, and Dawn Song. Assessing generalization in deep reinforcement learning, 2019.
>
> ---
>
> We sincerely hope the above responses have clarified your doubts regarding the paper and strengthened your confidence in our paper. If not, give us a holler and we are more than happy to further clarify.

---

> > ### Comment · Reviewer_8Wih · 2024-11-26
> >
> > I thank the authors for their careful response.
> >
> > However, I have additional observations regarding the hypervolume metric discussion:
> > - In the setting of linear utility, the hypervolume metric can be increased by adding solutions in concave regions of the PF, which do not result in an increase of utility. Hence, hypervolume would be misleading in the context of linear utility.
> > - The hypervolume metric is difficult to interpret and does not give information about the shape of the PF or which preferences/trade-offs the agent is able to solve.
> > - Computing hypervolume is NP-Hard, and is infeasible to compute in problems with many objectives (e.g. 10 objectives).
> > - While the authors addressed the problem of different reward magnitudes by normalizing the PF, I believe introducing a normalized metric of expected utility (and other metrics) are necessary for better evaluation of MORL algorithms due to the limitations of hypervolume.
> >
> > Similar observations regarding utility-based metrics have been made in the MORL literature, e.g., see the survey of Hayes et al. 2022.

---

> ### Author Response · Authors · 2024-11-27
> **Response to Reviewer 8Wih (Discussing EUM)**
>
> We sincerely thank you for your prompt engagement during this discussion phase. Before addressing your points, we hope to clarify a few aspects of the Expected Utility Metric (EUM) and its role in our work.
>
> * **Generality of the evaluation measure**
>
> First, this is mentioned in Hayes et al. (2022): *"This metric does however require a good prior over possible scalarisation functions in order to meaningfully evaluate a given solution set"*. This highlights a key limitation of EUM: when building a MORL generalization benchmark, we often lack prior knowledge of the true utility functions governing the domain. In fact, it is totally plausible that each context within a domain has its own distinct utility function distribution. As the first work in this area, our primary focus is to introduce a general evaluation metric that can assess generalizability across any MORL domain, without assuming prior knowledge of the underlying utility functions. However, your concerns about Hypervolume and EUM are totally valid, see next point $\rightarrow$
>
> * **Can expected utility metric be used?**
>
> Yes, *if* a well-informed prior over utility functions is available, EUM is indeed a valuable tool for evaluation. In response to this point, we have added a new section, Appendix D.1, which provides a detailed discussion of how EUM can be applied for MORL generalization evaluations. Additionally, we introduce the *Expected Utility Generalization Ratio* (EUGR), a measure motivated by NHGR, to enable meaningful comparisons when utility priors are available. This section also includes plots illustrating results obtained using EUM and EUGR. We hope you can take a moment to review this addition, it should address your concerns comprehensively.
>
> ---
>
> > In the setting of linear utility, the hypervolume metric can be increased by adding solutions in concave regions of the PF, which do not result in an increase of utility.
>
> You are correct, and we agree. In the very specific case where the stakeholders' preferences are known to be linear across all contexts, it may be an overkill to maximize hypervolume. As such, in these cases, one should use EUGR. Appendix D.1 and our codebase provide detailed steps for applying EUGR.
>
> > The hypervolume metric is difficult to interpret and does not give information about the shape of the PF or which preferences/trade-offs the agent is able to solve.
>
> While hypervolume does not explicitly reveal the shape of the PF, it is important to clarify that expected utility also does not inherently provide this information. As discussed earlier, expected utility relies heavily on prior knowledge of the underlying utility functions. It is this prior knowledge—**not the metric itself**—that defines the apparent shape of the PF. Without such prior information, assuming a utility function (e.g. linear scalarization) only reduces the problem scope and provides zero insight into the shape or structure of the PF.
>
> > While the authors addressed the problem of different reward magnitudes by normalizing the PF, I believe introducing a normalized metric of expected utility (and other metrics) are necessary for better evaluation of MORL algorithms due to the limitations of hypervolume.
>
> To conclude, we believe that there is significant alignment between your feedback and our perspective. In fact, we are grateful for your comments, as they highlighted an area of the paper that needed further discussion. The previous version of our paper prioritized a **general** approach to MORL generalization evaluations but lacked discussions on rare cases involving **specific predefined utility functions**.
>
> In this new draft, we retain the general evaluation method, i.e. NHGR, in the main body, while expanding the discussion in Appendix D.1 to address specific scenarios where utility functions are predefined, i.e. EUGR. We hope you find these additions make the paper more comprehensive and balanced. Would you agree that this updated version better addresses the breadth of MORL generalization evaluation needs?
>
> ---
>
> We sincerely hope the above responses have improved your confidence in our paper. As always, we are happy to clarify any further questions you have! 😊

---

> > ### Comment · Reviewer_8Wih · 2024-11-27
> >
> > I thank the authors for their careful response and for taking into account the feedback of the reviewers. I have increased my score accordingly.
> >
> > I would like to further discuss the problem formulation in Definition 3. One concern I have is that it assumes that *one single policy* should be optimal w.r.t. to a distribution of contexts. The authors have discussed the Principle of Unchanged Optimality appropriately, but I wonder if an alternative definition could also be appropriate. For instance, let's say the goal is to learn a set of policies that contains an optimal policy for each context separately:
> >
> > $\pi^*_{w,c} = \arg\max_{\pi} u_w(v^{\pi}_{c}), \forall c$
> >
> > Another option could also be to define policies that are conditioned on a context variable $\pi(s,c)$. This type of context-dependent policy has been explored in single-objective RL, and I believe it would be adequate to tackle generalization in MORL, too.
> >
> > The authors state that: "The problem, however, lies with the immaturity of current MORL algorithms for generalization which largely assume Markovian policies". I am not sure this is a valid claim, otherwise Definition 3 would have to explicitly mention that the optimal policies for this problem are non-Markovian. For this reason, I would suggest adding an alternative formulation with Markovian context-conditioned policies, since future algorithms would probably follow this approach.

---

> ### Author Response · Authors · 2024-11-30
> **Thank you Reviewer 8Wih**
>
> Thank you so much for actively engaging in the discussions and for providing valuable and constructive feedback on our paper. We feel fortunate to have a reviewer as invested as you.
>
> Your question is intriguing. It seems your interpretation of Definition 3 and ours are not fundamentally at odds but rather reflect different perspectives. In your interpretation, the policy explicitly conditions on the context, whereas in our definition, it does not. While the contextual MDP framework does not inherently restrict whether the context is hidden or observable, generalization discussions are typically interested in hidden-context scenarios. This is because, when the context can be directly conditioned upon, the contextual MDP problem essentially reduces to a *"universal MDP"*, where the context becomes part of the state space. In real-world generalization scenarios, however, the context is often unobservable during testing, making context-conditioned policies less relevant or practical for addressing the key challenges of generalization.
>
> To our best knowledge, we have not seen any context-dependent policy formulations in RL generalization literature. We do, however, see it in other research areas such as multi-task RL/meta-RL (condition on the task ID) or contextual bandit problems. Definition 3 does not restrict on whether the optimal policies are Markovian or non-Markovian, or whether the context can be conditioned upon. We believe it is important to present this general perspective in the main body of the paper.
>
> We are unable to update the manuscript on OpenReview at this stage. However, as per your advice, we have added a brief discussion in the appendix of our revised manuscript that discusses specific scenarios where context-conditioned policies may be applied and their distinctions from generalization-focused problems. We have also added a line in the main body to explicitly highlight to future MORL generalization work to consider non-Markovian methods to tackle problems where context can be inferred across multiple time steps. Once again, we sincerely thank you for your insightful feedback and for inspiring us to refine the paper further. Feel free to let us know if you have any further questions. Your confidence in our work means a lot to us, and it has been an absolute pleasure engaging in these discussions with you!

---

> ### Author Response · Authors · 2024-12-02
> **Checking in**
>
> We hope this message finds you well. As the discussion period nears it end, we want to make sure that we have addressed any remaining concerns you may have. Our goal is to ensure that our work serves as a strong foundation for future research into generalization in MORL. Your thoughtful insights have been crucial in shaping the trajectory of our work, and we would like to sincerely thank you for your valuable contributions.

---

### Author Response · Authors · 2024-11-27
**Summary of discussions**

We extend our heartfelt thanks to the reviewers who engaged with us throughout this discussion period. As the discussion draws to a close, we wish to take this opportunity to reiterate the contributions of our work and summarize the discussions with the reviewers.

&nbsp;

---

&nbsp;

## Primary Contributions of our Work

Real world problems are often inherently multi-objective and require generalization over various scenarios. However, prior to our work, **no research had addressed generalization in Multi-Objective Reinforcement Learning (MORL)**. Existing MORL research is limited to static environments and present RL generalization literature only considered scalar rewards. This paper stands as the pioneering effort in this critical intersection of RL research. All reviewers have acknowledged the importance of this direction, and its potential to advance the broader field of RL.

As the first work in this nascent area of research, there are many potential areas that we could possibly have looked into. Ultimately, we decided it would be most meaningful to focus our efforts on contributions that would effectively lay a strong foundation for future studies in MORL generalization. Considering the lack of prior research in this area, we are genuinely proud of the work our team has established.

To best advance this field, we have provided:
1. **Formalisms:** a *general* framework to discuss and evaluate generalization in MORL using theoretically motivated metrics (NHGR and EUGR),
2. **Benchmarks:** a novel benchmark comprising six diverse domains with rich environment configurations carefully designed to tackle generalization involving state-space, dynamics, and reward variations (see Appendix F),
3. **Datasets and Baselines:** extensive evaluations of current state-of-the-art (SOTA) algorithms with open-sourced dataset, and post-hoc analyses of the results,
4. **Extensive Discussions:** detailed discussions on the failure modes of existing SOTA methods, and possible future directions of MORL generalization to improve them

Most importantly, we provide **open-source software** to streamline MORL generalization training and evaluation across these six domains, along with a raw dataset from over 1,000 GPU hours of evaluations involving eight SOTA MORL algorithms. Our contributions lays the necessary groundwork (formalism, software tools, benchmarks, and insights) for accelerating future research on generalization in multi-objective domains, ultimately pushing the boundaries of what RL agents can accomplish in complex, real-world scenarios. Our contributions also align significantly with the **Datasets & Benchmarks** agendas of ICLR, an important pillar for advancing machine learning research.

---

&nbsp;

## The Path Forward

Regardless of this conference submission’s outcome, we are confident that the software tools, datasets, benchmark, and insights we have established in this paper will be of value to the RL community. This research direction will likely attract significant attention in the years ahead, especially given the growing interest in developing real-world agents that are aligned with humans' pluralistic needs. As a pioneering effort, our work opens the door to numerous exciting avenues for further exploration. We look forward to continue advancing this direction through open-source contributions and future research.

To interested readers and researchers, here are key ways we can collectively advance this field:

1. **Open-sourced Software:** We are proud to share that our code is freely available at **[https://anonymous.4open.science/r/morl-generalization](https://anonymous.4open.science/r/morl-generalization)**. We invite researchers to use our software pipeline and benchmark to begin investigations into MORL generalization.
2. **Dataset:** All results used in our paper are included as CSVs within our open-source code repository. You can leverage our existing dataset for baseline comparisons, eliminating the need to rerun baseline experiments.  We are unable to release the full raw evaluation dataset currently as it is tied to a non-anonymous *Weights & Biases* account, but will do so after the conference decision.
3. **Useful Insights:** We refer motivated readers to the appendix section of our paper, which delves into detailed discussions on the limitations of current algorithms and actionable insights for advancing algorithmic improvements to tackle MORL generalization. There are many potential areas to tap into.

We are committed to ensuring that our work serves as a solid foundation for future research into generalization in MORL. If you have any questions regarding our work, please do not hesitate to reach out. We hope our work spurs greater recognition of the importance of multi-objective reward structures for RL generalization and we’re eager to see how future research uses our software and benchmark to advance MORL generalization! 😃

---

> ### Author Response · Authors · 2024-11-27
> **Addressing reviewer comments (part 1/2)**
>
> We have meticulously addressed the reviewers' comments throughout this discussion period. All reviewers have also acknowledged the value of our pioneering work to the broader RL field in their reviews. We would like to **specially thank Reviewers 8Wih and nVsL** for actively engaging in discussions and for providing constructive feedback that have improved our paper significantly. Below, we provide a summary of the discussions:
>
> ---
>
> ### Justifying NHGR
>
> Some reviewers had questions about the *Normalized Hypervolume Generalization Ratio* (NHGR) metric. However, we have explained NHGR’s favorable properties, justifying its retention as a **general and fair evaluation metric** for MORL generalization.
>
> Here, we briefly reiterate the benefits of the NHGR metric (for detailed explanations, please refer to Appendix D.3). Standard hypervolume is scale-biased and depends on arbitrary reference points. While normalized hypervolume ($HV_{\text{norm}}$) resolves these, it remains biased as it does not account for differences in maximally achievable $HV_{\text{norm}}$ across environments. This unfairly favor agents that perform well only in environments with convex-like Pareto fronts and higher maximal-achievable hypervolumes rather than those who divide their learning to improve in every environment, even those with more concave fronts. Such bias contradicts the **motivations of generalization in RL**. NHGR addresses this discrepancies by evaluating generalist agents' performance as a ratio of approximated maximal hypervolumes, ensuring fair consideration of all evaluation contexts. NHGR is based on the **hyperarea measure** in multi-objective optimization, which has many favorable mathematical properties.
>
> In single-objective RL generalization, arbitrary estimates are commonly used to determine “success rates” or “optimal gaps”. However, arbitrary estimates often fail to accurately measure the true competencies of RL algorithms. NHGR uses specialist performances to determine these thresholds, providing more reliable comparisons. Indeed, in future benchmarks where specialists can't perform well, it might lead to inaccurate NHGR approximations. We would like to note that for the 6 domains in our benchmark, our specialists can perform well on them. As such, using NHGR has significant benefits for fair evaluations compared to standard hypervolume, as explained earlier. Note that both hypervolume and NHGR are also native to our software pipeline and are tabulated during training. We provide **both results in our revised manuscript** for more flexible future evaluations.
>
> ---
>
> ### Utility-based evaluations: EUGR
>
> Some reviewers also suggested the use of *expected utility metric* (EUM) over hypervolume. However, it is important to emphasize a key limitation of EUM, as noted in Hayes et al. (2022): *"This metric does however require a good prior over possible scalarisation functions..."*. In most MORL generalization benchmarks, the true utility functions governing the domain are often unknown.  As the pioneering work in this field, our primary focus is to introduce a **general evaluation metric** that can assess generalizability across any MORL domain, regardless of the availability of utility priors. As such, hypervolume-based metrics, i.e. our proposed NHGR, is more fitting than EUM.
>
> That said, we are grateful to reviewers for highlighting an area of the paper that could use further discussion. Indeed, *in specific cases* where a well-informed prior over utility functions is available, EUM is a valuable tool for evaluation. In response to this point, we have added a new section in Appendix D, which discusses the use of utility-based metrics for MORL generalization evaluations. In that section, we introduce the *Expected Utility Generalization Ratio* (EUGR), a measure motivated by NHGR, to enable meaningful comparisons when utility priors are available. This section also includes plots illustrating results obtained using EUM and EUGR. In this new draft, we retain the general evaluation method, i.e. NHGR, in the main body, while expanding the discussion in Appendix D to address specific scenarios where utility functions are predefined, i.e. EUGR.
>
> ---
>
> Ultimately, as a pioneering work, our goal is not to impose rigid constraints on future research but to **establish flexible and well-justified options**. The choice of performance metric has been an **ongoing debate in multi-objective optimization for decades**. This is because fundamentally, measuring the quality of Pareto fronts is more complex than unidimensional evaluations, and **no single metric can suffice as the ultimate solution**. We therefore **provide evaluations using multiple metrics**, i.e. hypervolume, NHGR, EUM and EUGR, throughout our revised manuscript, This provides more informative and flexible options for evaluations. Once again, we must emphasise that our software tool is flexible to accommodate the use of standard/new metrics in future research.

---

> ### Author Response · Authors · 2024-11-27
> **Addressing reviewer comments (part 2/2)**
>
> ### Reiterating Contributions of our work
>
> Lastly, reviewers raised some comments which we felt misunderstood/undervalued the contributions of our work. We summarise these statements and our responses to them below:
>
> ---
>
> > The theoretical contributions are limited to the definition of a contextual MOMDP, which follows directly from previously studied formalisms of MOMDP and contextual MDPs
>
> This paper was not meant to be a theory paper, and we apologize to reviewers if we gave a different impression. Our paper seeks to establish **general formalisms** which future research can reference and build upon. Our goal here is not to provide restrictive assumptions about the MOC-MDP such as Lipschitz smoothness in variations of parameters which is canonical in single-objective RL generalization literature. Theoretical studies involving such assumptions would naturally follow our work and form a complementary body of literature in this research direction.
>
> ---
>
> > The proposed testbed appears to be a straightforward extension of MORL environments to generalization with limited variations
>
> We must emphasize that is not true. To create our benchmark, it involved **substantial engineering efforts and careful design choices to support meaningful generalization studies**. This took our team many months of effort to establish. For instance, working with the Mujoco engine requires understanding of its XML configurations to implement context variations. Designing these benchmarks also require many trial-and-error runs to validate the environments are "learnable." For example, in MO-Hopper and MO-LunarLander, setting the friction/gravity values inappropriately would render the task unlearnable. MO-SuperMarioBros, previously limited to stage 1-1, was expanded to encompass 32 distinct stages. Additionally, MO-LavaGrid is an entirely new environment we designed to test MORL generalization across reward functions.
>
> We also made sure that our benchmark captures the complexity of real-world generalization challenges (detailed discussions in Section 5 and Appendix F.1). Specifically, we deliberately ensured coverage of **state-space, dynamics, and reward-function variations** in our domains, each targeting a distinct dimension of generalization, as outlined in [1]. Also, [1] notes that *"...dynamics variation is only tackled in limited settings and reward-function variation is very under-studied. These stronger forms of variation are still likely to appear in real-world scenarios, and hence should be the focus of future research"*. By introducing dynamics-based and reward-function variations, our benchmark addresses critical gaps in current RL generalization research, while additionally incorporating multi-objective aspects to better align with real-world complexities. The poor performance of current MORL methods also highlights the benchmark’s potential to become a lasting standard for future research in MORL generalization. These contributions signify the relevance and utility of our benchmark for driving RL research.
>
> [1] Robert Kirk, Amy Zhang, Edward Grefenstette, and Tim Rockt¨aschel. A survey of zero-shot generalisation in deep reinforcement learning. J. Artif. Int. Res., 76, May 2023
>
> ---
>
> > The overall contribution and novelty of this paper remain incomplete. While the paper highlights a promising research direction, it lacks a concrete solution for addressing the problem of MORL generalization. To enhance the paper’s completeness, the authors should propose and integrate their own method to tackle MORL generalization effectively.
>
> Some reviewers commented that our paper "lacks completeness" because we did not propose any new algorithms. We respectfully disagree with this statement. While proposing new algorithms is undeniably valuable, we must emphasize that this paper focuses on establishing the foundational groundwork for MORL generalization. As such, our contributions already include extensive coverage of necessary formalisms, software tools, datasets, benchmarks, and extensive post-hoc discussions—all of which are critical for pioneering this field.
>
> The notion that introducing a new algorithm is necessary for the paper to be considered *"complete"* reflects a narrow interpretation of what constitutes a meaningful contribution to research and overlooks the significant groundwork we have laid. Critical pillars of machine learning research, such as theory work, blue-sky ideas and **Datasets & Benchmark** that our paper is aligned with are equally as important as *“pushing the state-of-the-art algorithmic publications”*. We hope the reviewers can empathize with the constraints of a conference-length publication.  The development of a novel algorithm, along with the theoretical and empirical analyses it would require, would constitute the scope of an entirely separate paper succeeding ours. In fact, Appendix C of our paper exactly discusses how these algorithmic innovations can happen!

---

### Meta-Review · Area_Chair_AQc7 · 2024-12-21

**Metareview:**

The primary contribution of this paper is to specify a problem setting, generalization to unobserved contexts in multi-objective reinforcement learning, introduce benchmark environments for this setting, and show that existing methods cannot solve these problems. This paper accomplishes its goal. The reviewers did not note any major errors in the paper to warrant rejects. So, I recommend this paper for acceptance.

**Additional Comments On Reviewer Discussion:**

There were discussions between reviewers and authors and most points of contention were resolved. There was nothing major left to object to accepting the paper.

---

### Decision · Program_Chairs · 2025-01-22

Accept (Poster)